# LATMiX: Learnable Affine Transformations for
# Microscaling Quantization of LLMs

Ofir Gordon [1]   Lior Dikstein [1]   Arnon Netzer [1]   Idan Achituve [*1]   Hai Victor Habi [*1]

## Abstract

Post-training quantization (PTQ) is a widely used approach for reducing the memory and compute costs of large language models (LLMs). Recent studies have shown that applying invertible transformations to activations can significantly improve quantization robustness by reducing activation outliers; however, existing approaches are largely restricted to rotation or Hadamard-based transformations. Moreover, most studies focused primarily on traditional quantization schemes, whereas modern hardware increasingly supports the microscaling (MX) data format. Attempts to combine both showed severe performance degradation, leading prior work to introduce assumptions on the transformations. In this work, we take a complementary perspective. First, we provide a theoretical analysis of transformations under MX quantization by deriving a bound on the quantization error. Our analysis emphasizes the importance of accounting for both the activation distribution and the underlying quantization structure. Building on this analysis, we propose LATMiX, a method that generalizes outlier reduction to learnable invertible affine transformations optimized using standard deep learning tools. Experiments show consistent improvements in average accuracy for MX low-bit quantization over strong baselines on a wide range of zero-shot benchmarks, across multiple model sizes.

## 1. Introduction

Large language models (LLMs) have become a foundational component in a wide range of applications, including natural language understanding, code generation, reasoning, and multimodal systems (Brown et al., 2020; Chen,

2021; Alayrac et al., 2022; Wei et al., 2022a). Recent years have witnessed a rapid increase in both the scale and capability of these models, leading to substantial gains in performance and generalization ability (Kaplan et al., 2020; Hoffmann et al., 2022; Zhao et al., 2023). However, this scaling trend comes at a significant computational and memory cost, making efficient deployment increasingly challenging. Post-training quantization (PTQ) (Gholami et al., 2022) has therefore emerged as a key technique for reducing inference latency, memory footprint, and energy consumption while preserving model accuracy, without requiring expensive retraining (Lin et al., 2016; Nagel et al., 2020). As a result, a rich body of work has proposed diverse PTQ methods for LLMs, demonstrating that carefully designed quantization strategies can substantially lower deployment costs with mild performance degradation (Dettmers et al., 2022; Frantar et al., 2023; Xiao et al., 2023).

However, existing PTQ methods for LLMs often struggle in severely resource-constrained settings that require low bit widths, such as 4-bit quantization, where accuracy degradation becomes more pronounced. A key factor limiting performance in this regime is the presence of activation outliers, which dominate quantization error and prevent effective low-precision representation. Motivated by the intuition that redistributing energy more evenly across dimensions improves quantization robustness, several recent works mitigate activation outliers using invertible transformations, most notably rotation-based methods (Chee et al., 2023; Ashkboos et al., 2024c; Liu et al., 2025).

In parallel, new data formats have emerged, such as the microscaling (MX) format, introduced by the Open Compute Project (Rouhani et al., 2023b). The MX format is specifically designed to better accommodate the numerical characteristics of large models and is endorsed by Microsoft, AMD, Arm, Intel, Meta, and NVIDIA. The key idea behind MX is to partition the tensors into blocks, where each block is assigned its own scaling factor that can be dynamically selected (Rouhani et al., 2023a). This allows finer-grained control over the quantization error. This fine-grained scaling enables both efficient training (Chmiel et al., 2025) and inference (Lee et al., 2025a) using MX quantization.

However, recent studies have shown that naively combin-

*Equal contribution  [1]Arm, Israel. Correspondence to: Hai Victor Habi <haivictor.habi@arm.com>.

*Proceedings of the 43$^{rd}$ International Conference on Machine Learning*, Seoul, South Korea. PMLR 306, 2026. Copyright 2026 by the author(s).

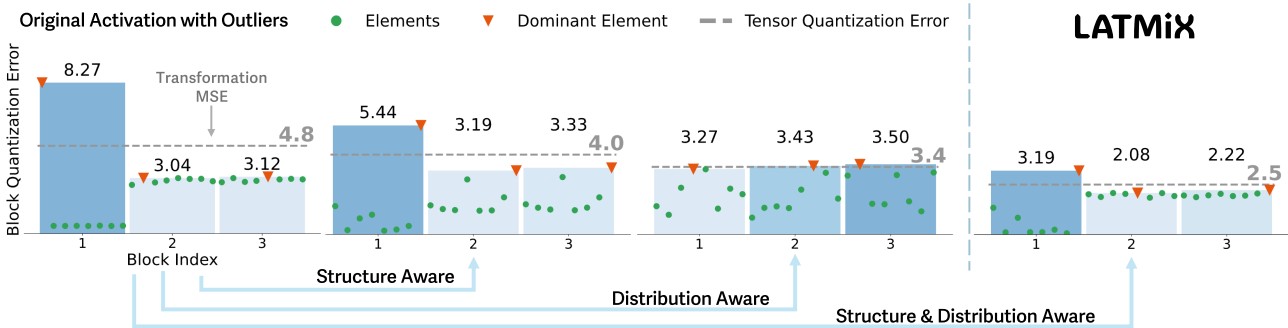

*Figure 1.* LATMiX takes into account both the MX block structure and the distribution of features to diffuse outliers. In the figure, energy is distributed both within the block and among blocks to obtain lower quantization error.

ing MX quantization with global rotation- or Hadamard-based transformations leads to severe performance degradation ([Egiazarian et al., 2025](); [Shao et al., 2025b]()). As a workaround, these studies suggest applying such transformations independently within each MX block using block-diagonal rotation matrices. While this design choice avoids the immediate mismatch between global transformations and block-wise scaling, it fundamentally restricts the transformation to operate on small, isolated subspaces, preventing cross-block redistribution of activation mass. As a result, dominant outliers cannot be effectively diffused across the full tensor, limiting outlier suppression and leading to suboptimal quantization accuracy in low-bit regimes.

In this study, we address this gap by first presenting a theoretical and numerical analysis showing that quantization error depends on both the *feature distribution* and the *MX block structure*, with block-diagonal matrices representing only a special case. To more generally account for both factors, we propose LATMiX. Specifically, we define the class of admissible transformations as all invertible affine transformations, without imposing independence assumptions. The transformations are learned by optimizing free-form parameters corresponding to LU and QR decompositions of general invertible matrices, using a distillation loss and a volume-preserving regularization. The former encourages the predictions of the transformed quantized model to match those of the full-precision model, while the latter maintains invertibility throughout optimization. Since the learned transformations can be folded into the linear layers, they incur negligible inference cost. We validate this claim through throughput measurements on real hardware. This enables a substantially richer family of transformations, allowing more effective redistribution of activation mass and improved outlier mitigation.

To summarize, in this study we make the following novel contributions: (1) We conduct a theoretical and numerical study of MX quantization, emphasizing the importance of both the block structure and feature distribution; (2) We propose LATMiX, a method for learning affine transformations

that can be folded into weight matrices based on LU and QR parameterizations; (3) We present experimental results on seven zero-shot reasoning benchmarks and WikiText2 using various LLMs. Overall, LATMiX shows consistent improvements over leading baselines. In the spirit of research reproducibility, the code for this work is available at https://github.com/arm-research/AAIR-LATMiX

## 2. Background & Notation

We use bold lowercase letters to denote vectors (e.g., $\boldsymbol{x}$) and bold uppercase letters to denote matrices (e.g., $\mathbf{W}$). For notational convenience, and without loss of generality, when not written explicitly we assume that vectors lie in $\mathbb{R}^d$ and matrices in $\mathbb{R}^{d \times d}$. Throughout the paper, $\boldsymbol{x}$ is used as a generic notation for neural network activations.

### 2.1. Microscaling (MX) Quantization

Here we briefly outline MX quantization ([Rouhani et al., 2023b]()), focusing on block-wise power-of-two dynamic scaling commonly used in conjunction with low-precision element formats such as FP4. In standard tensor-wise quantization, using one shared scale for an entire tensor can be suboptimal when values vary widely in magnitude. MX quantization addresses this by partitioning the tensor into small blocks, each of which is assigned its own scale factor rather than sharing a single global scale. More formally, let $\boldsymbol{x} \in \mathbb{R}^d$ be the vector to be quantized, $B$ the MX block size, and $N_B = d/B$ the number of blocks (assuming $B$ divides $d$). The quantization of an element $\boldsymbol{x}_j$ belonging to block $i \in \{0, \ldots, N_B - 1\}$ is defined as:

$$\text{Compute scale:} \quad s_i = 2^{\left\lfloor \log_2 \left( \max_{j \in \mathcal{I}_i} |\boldsymbol{x}_j| \right) \right\rfloor - r_{\max}},$$

$$\text{Quantize:} \quad Q\left(\boldsymbol{x}\right)_j = s_i Q_e \left( \frac{\boldsymbol{x}_j}{s_i} \right), \quad (1)$$

where $\mathcal{I}_i = [i \cdot B, \ldots, (i+1) \cdot B - 1]$ denotes the index set of the $i^{th}$ block, $Q_e(\cdot)$ is the element-wise quantizer cor-

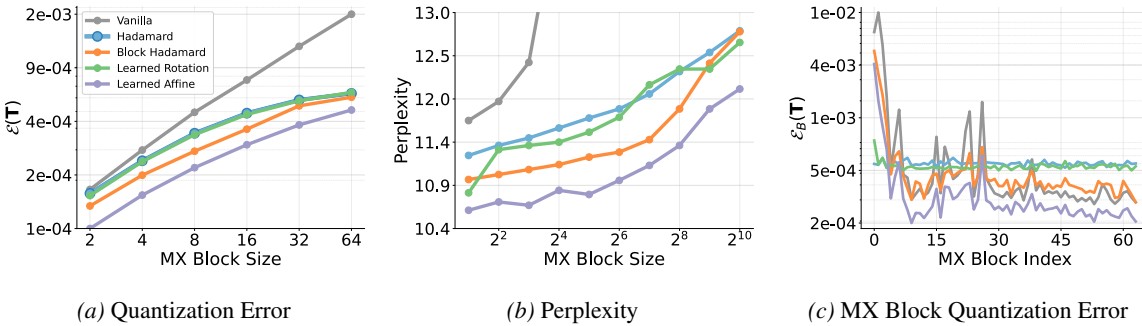

*Figure 2.* Analysis of various transformation types: (1) Vanilla: no transformation applied; (2) Hadamard: Full Hadamard transform; (3) Block Hadamard: a block-diagonal matrix in which each block corresponds to an MX block with an Hadamard matrix; (4) a learned rotation matrix; and (5) a learned affine transformation that minimizes the objective in Eq. (2). In Fig. 2a, the Hadamard and learned rotation curves are superimposed.

responding to the chosen low-precision format (e.g., FP8, INT4, FP4), and $r_{\max}$ denotes the maximum exponent representable in that format. Throughout this work, we denote by $Q(\boldsymbol{x})$ the MX quantization operator defined in Eq. (1) applied on to the entire tensor $\boldsymbol{x}$.

## 2.2. Outlier Reduction Using Rotation Matrices

In the literature, it has been observed that rotating the activations of a neural network (NN) prior to quantization helps reduce outliers, thereby lowering the quantization error (Tseng et al., 2024; Ashkboos et al., 2024c; Liu et al., 2025). Rotation transformations, as a subset of invertible linear transformations, are typically chosen in order to preserve computational invariance (Ashkboos et al., 2024a), namely, that the network with rotated activations remains functionally equivalent to the original network. This invariance is achieved by injecting the inverse transformation into the network after rotating the activations. More formally, consider a linear layer with activation $\boldsymbol{x}$, a weight matrix $\mathbf{W}$, and a rotation matrix $\mathbf{A}$, then, $\boldsymbol{x}^{\top}\mathbf{W} = (\mathbf{A}\boldsymbol{x})^{\top}(\mathbf{A}^{-\top}\mathbf{W})$.

Recently, it was proposed in (Liu et al., 2025) to learn the rotation matrices from a small calibration set, departing from previous approaches which set it in advance to some value. During the optimization process, weights are kept in full precision, while only the rotated activations are quantized. After learning the rotation matrices, the corresponding transformed weights are quantized as well. However, the approach proposed in (Liu et al., 2025) is restricted to rotation matrices only, and the optimization, which is performed directly on the Stiefel manifold, can be less amenable to common deep learning optimization techniques (Li et al., 2020; Huang et al., 2023). Furthermore, recent studies (Egiazarian et al., 2025; Shao et al., 2025b) have shown that when MX quantization is applied to rotated activations, the outlier suppression can be severely harmed, resulting in substantial performance degradation. Hence, these studies suggested applying the rotations separately per MX block.

## 3. Method

We first conduct a theoretical and numerical study of quantization error under various transformations in MX quantization. Our findings show that both the activation distribution and the MX block structure play a crucial role in designing transformations that effectively reduce quantization error. Building on this analysis, we introduce LATMiX, a method that learns transformations from the class of invertible affine transformations. An illustration of the proposed approach is shown in Fig. 1.

## 3.1. Theoretical & Numerical Analysis

Here, we provide a theoretical and numerical analysis of the transformation effect on the microscaling quantization error. We begin by defining the affine transformation $\mathbf{T}$, its inverse and the MSE error under MX quantization.

**Definition 3.1** (General Affine Transformation). Let $\mathbf{A} \in \mathbb{R}^{d \times d}$ be an invertible matrix and $\boldsymbol{v}$ be a translation vector. Then an affine transformation is defined as $\mathbf{T}(\boldsymbol{x}) = \mathbf{A}\boldsymbol{x} + \boldsymbol{v}$ and its inverse $\mathbf{T}^{-1}(\boldsymbol{y}) = \mathbf{A}^{-1}\boldsymbol{y} - \mathbf{A}^{-1}\boldsymbol{v}$.

To examine various types of transformations, we employ the Mean Squared Error (MSE), which approximates changes in the objective function induced by perturbations in weights and activations (Nagel et al., 2020; Gordon et al., 2024). This approach relies on the following assumptions: (i) the model is well trained, so the gradients are close to zero; (ii) a second-order Taylor expansion is used; and (iii) the layers are assumed to be independent of one another, resulting in a block-diagonal Hessian. We would like to stress though that we use the MSE here solely to analyze different transformations and these assumptions are not required by LATMiX.

**Definition 3.2** (Transformation Mean Squared Error). The Transformation Mean Squared Error for $\mathbf{T}$ is defined as:

$$\mathcal{E}(\mathbf{T}) \triangleq \tfrac{1}{d}\mathbb{E}\left[||\boldsymbol{x} - \mathbf{T}^{-1}(Q(\mathbf{T}(\boldsymbol{x})))||_2^2\right], \quad (2)$$

where $\|\cdot\|_2$ denotes the Euclidean norm of a vector.

**Theoretical Analysis**. We first derive an upper bound for the quantization error defined in Eq. 2 for general distributions, and then present its concrete form in the case of $\psi_\alpha$ distributions , allowing us to draw practical insights. The class of $\psi_\alpha$ distributions is broad and generalized, encompassing both sub-Gaussian and sub-Exponential distributions; in particular, any centered random variable with bounded support is sub-Gaussian (Vershynin, 2026).

As our interest lies in extreme-value behavior, $\psi_\alpha$ distributions are well suited due to their Gaussian-dominated tails.

**Theorem 3.3** (MX Quantization Error). *Assume that $\boldsymbol{x}$ is a continuous random vector, $\mathbf{T}$ is an affine transformation and Q is the quantization of MX as defined in Eq. (1). Then, under regularity assumptions on $\boldsymbol{x}$,*

$$\mathcal{E}\left(\mathbf{T}\right) \lesssim \frac{\|\mathbf{A}^{-1}\|_\sigma^2}{N_B} \sum_{i=1}^{N_B} M_i, \quad where$$

$$M_i \triangleq \mathbb{E}\left[\left(\max_{j \in \mathcal{I}_i}\left|\left[\mathbf{T}\left(\boldsymbol{x}\right)\right]_j\right|\right)^2\right]. \tag{3}$$

*Here, $f(x) \lesssim g(x)$ denotes that $f(x)$ is less than $g(x)$ up to a fixed multiplicative constant, and $\|\cdot\|_\sigma$ denotes the spectral norm. Furthermore, if we assume that $\boldsymbol{x}$ is a $\psi_\alpha$ random vector with mean $\boldsymbol{\mu}$ then,*

$$M_i \lesssim \left(\mathbf{T}_{max}(\boldsymbol{\mu}) + h_\alpha\left(B\right) \max_{j \in \mathcal{I}_i}\left\|\left[\hat{\mathbf{T}}\left(\boldsymbol{x}\right)\right]_j\right\|_{\psi_\alpha}\right)^2. \tag{4}$$

*Where $\|z\|_{\psi_\alpha} \triangleq \inf\left\{K > 0 : \mathbb{E}\left[\exp\left(\frac{z^\alpha}{K^\alpha}\right)\right] \leq 2\right\}$ is the Orlicz norm of the random variable z, $h_\alpha\left(B\right)$ is a constant depending on the block size B and the decay of the tail $\alpha$, $\mathbf{T}_{max}(\boldsymbol{x}) \triangleq \max_{j \in \mathcal{I}_i}\left|\left[\mathbf{T}\left(\boldsymbol{x}\right)\right]_j\right|$, and $\left[\hat{\mathbf{T}}\left(\boldsymbol{x}\right)\right]_j \triangleq \left[\mathbf{T}\left(\boldsymbol{x}\right) - \mathbf{T}\left(\boldsymbol{\mu}\right)\right]_j$.*

From Theorem 3.3 (see the proof in Appendix A), we see that the MSE is determined by two factors: (1) the norm $\|\mathbf{A}^{-1}\|_\sigma^2$, and (2) the average, across MX blocks, of the expected maximum absolute channel value within each block. Reducing either term leads to a lower transformation MSE. However, these factors can be at odds. Reducing the first term, by increasing the smallest singular value of $\mathbf{A}$, may increase $\sum_{i=1}^{N_B} M_i$, while transformations that reduce this sum can simultaneously reduce the smallest singular value, thereby increasing $\|\mathbf{A}^{-1}\|_\sigma^2$. The existence of this tradeoff hints that learning the transformation is required to navigate and balance between the terms.

Furthermore, Eq. (3) provides key insights into the design of the transformation $\mathbf{T}$. In particular, choosing $\mathbf{A}$ as a random

invertible matrix can be detrimental, as indiscriminate channel mixing may increase quantization error. To illustrate, consider a length-four vector with a Dirac delta distribution centered at $[10, 1, 0.5, 0.5]$, and an MX block size of $B = 2$. Applying the Walsh–Hadamard matrix $\mathbf{T} = \mathbf{H}_4$ yields the rotated vector $[6, 4.5, 5, 4.5]$, which reduces the error in the first block but increases it in the second block. Motivated by such behavior, prior work (Egiazarian et al., 2025; Shao et al., 2025b) advocates the use of block-diagonal transformations. This choice is supported by Eq. (3): owing to the additive structure of the bound, the error contribution of the $i^{\text{th}}$ MX block depends on $\|\mathbf{A}_i^{-1}\|_\sigma^2 M_i$, where $\mathbf{A}_i^{-1}$ denotes the submatrix corresponding to block $i$. Operating on each block independently can therefore reduce the per-block error and, consequently, the total error.

However, the block-diagonal approach implicitly assumes independence of the channels across blocks. As a result, even if the average per-block error decreases, the overall error may remain large. To address both sources of error, we argue for using full (non–block-diagonal) transformations that explicitly account for both the MX block structure and the channel distribution. One way to account for both the MX structure and the data distribution is to learn the transformation coupled with MX quantization on activations. From cardinality considerations, as $\mathrm{O}\left(d\right) \subset \mathrm{GL}\left(d\right) \subset \mathrm{Aff}\left(d\right)$, namely the group of orthogonal matrices is a subset of the general linear group which itself is a subset of the general affine group, then it immediately follows that $\min_{\mathbf{T} \in \mathrm{O}(d)} \mathcal{E}\left(\mathbf{T}\right) \geq \min_{\mathbf{T} \in \mathrm{GL}(d)} \mathcal{E}\left(\mathbf{T}\right) \geq \min_{\mathbf{T} \in \mathrm{Aff}(d)} \mathcal{E}\left(\mathbf{T}\right)$. Hence, we suggest to define the set of admissible transformations as the general affine group.

The benefit of adding the bias term, namely preferring $\mathrm{Aff}\left(d\right)$ over $\mathrm{GL}\left(d\right)$, is further corroborated when considering the $\psi_\alpha$ case (Eq. (4)). If $\mathbf{T}\left(\boldsymbol{\mu}\right) = \mathbf{0}$ before quantization, then $\mathbf{T}_{max}(\boldsymbol{\mu})$ is dropped from the bound on $M_i$. In principle, it can be obtained using the linearity of both the transformation and the expectation by setting the bias term according to $\mathbf{T}\left(\boldsymbol{\mu}\right) = \mathbf{A}\boldsymbol{\mu} + \boldsymbol{v} = \mathbf{0} \Rightarrow \boldsymbol{v} = -\mathbf{A}\boldsymbol{\mu}$.

We note that while prior studies (Egiazarian et al., 2025; Shao et al., 2025b) empirically investigated learning full transformations using the method proposed in Liu et al. (2025), which was not developed in the context of MX quantization, there are two important distinctions from our approach. First, the transformations in these studies are restricted to the set $\mathrm{O}(d)$, whereas we consider the broader class $\mathrm{Aff}(d)$. Second, the optimization procedure differs, as we optimize over free-form matrices and use a distillation loss instead of a cross-entropy loss. We discuss both points in detail in Section 3.2. These design choices can have a significant empirical impact, as demonstrated in Section 5, but first we analyze them numerically.

**Numerical Analysis**. To further illustrate the advantage of

learning affine transformations, we evaluate several common transformations in a numerical study in Figure 2 on features from Llama3.2-1B. Figure 2a shows the transformation MSE as a function of the MX block size, Fig. 2b further presents the perplexity for various block sizes showing favorable perplexity to the learned affine transformations approach, and Fig. 2c shows the block-wise quantization error defined as, $\mathcal{E}_B^i(\mathbf{T}) \triangleq \frac{1}{B} \sum_{j \in \mathcal{I}_i}([\boldsymbol{x} - \mathbf{T}^{-1}(Q(\mathbf{T}(\boldsymbol{x})))]_j)^2$. In consistent with previous studies (Egiazarian et al., 2025; Shao et al., 2025b), Figure 2a shows that for smaller block sizes block Hadamard transformation attains a lower error compared to the learned and non-learned full rotation transformations. Nevertheless, block Hadamard transformation under-performs learned affine transformation which achieve the smallest error on all block sizes. In addition, from Fig. 2c, compared to the vanilla transformation, full rotation-based transformations produce nearly uniform quantization error across blocks, thus increasing the overall error in most MX blocks. Block-Hadamard transformation mainly reduces the error in the dominant MX blocks (blocks with a high error) while increasing the error in the remaining MX blocks. And, learned affine transforms is the only methods that reduces the error across all MX blocks uniformly. In summary, across all the results, we observe that the learned affine transformations consistently outperform other approaches. This underscores the advantage of learning general affine transformations that consider both the MX block structure and the distribution of feature maps.

## 3.2. Learning General Affine Transformation

Our analysis in Section 3.1 reveals important design choices for constructing transformations that take into account both the MX data format and the features distribution. Most notably, defining the set of admissible functions to be invertible affine transformations. This is in contrast to previous studies that restrict the set of invertible transformations, usually to rotation matrices only. We now introduce LATMiX, a method that uses and learns *general* invertible affine transformations for reducing activation outliers. We stress here two important aspects of LATMiX: (1) Its inference runtime is similar to that of common rotation-based methods thanks to transformation folding [1]; (2) Although our approach was developed for MX data format, it can seamlessly be applied to other data formats.

We follow previous studies (Tseng et al., 2024; Ashkboos et al., 2024c; Liu et al., 2025) and define invertible transformations that can be absorbed into weight matrices, $\mathbf{T}_1$ and $\mathbf{T}_2$, and an online transformation $\mathbf{T}_3$ (see Fig. 5 for an illustration). Specifically, $\mathbf{T}_1$ acts globally on the input activations of all transformer blocks, while $\mathbf{T}_2$ and $\mathbf{T}_3$ act locally per transformer block on the activations of the scaled

---

[1]See Appendix C for further details.

dot-product layer in the attention layer and inside the feed-forward network, respectively. As in previous work (Liu et al., 2025; Egiazarian et al., 2025; Shao et al., 2025b) we learn $\mathbf{T}_1$ and $\mathbf{T}_2$ only; However, there are two important distinctions from prior studies, aside from the set of admissible invertible functions. First, we optimize free-form matrices, followed by deterministic transformations. As a result, common deep learning optimization algorithms can be used instead of constrained optimization on manifolds (Liu et al., 2025). Second, we soften the requirement of functional equivalence between the unquantized network and the transformed network, allowing them to be approximately equivalent by optimizing a distillation loss. We discuss both points in detail next.

**Parametrization of the transformations**. We start by introducing our approach to constructing invertible transformations. We present two parameterizations for $\mathbf{A}$; the first is inspired by (Kingma & Dhariwal, 2018), which uses LU decomposition, and the second uses QR decomposition. Starting with the former, $\mathbf{A}$ is decomposed according to:

$$\mathbf{A} = \mathbf{PL}\left(\mathbf{U} + \mathrm{diag}\left(\boldsymbol{s}\right)\right), \tag{5}$$

where $\mathrm{diag}(\cdot)$ means constructing a diagonal matrix with $\boldsymbol{s}$ on its diagonal, $\mathbf{P} \in \mathbb{R}^{d \times d}$ is a fixed permutation matrix, and the learned parameters are $\mathbf{L} \in \mathbb{R}^{d \times d}$ a lower unitriangular matrix, $\mathbf{U} \in \mathbb{R}^{d \times d}$ an upper triangular matrix with zeros on its diagonal, and the vector $\boldsymbol{s} \in \mathbb{R}^d$. Although the LU parametrization gives good empirical results, it is limited in the class of invertible transformations it can represent. Hence, we suggest a parametrization based on QR decomposition which spans the entire GL group. Concretely,

$$\mathbf{Q} = \exp\{\frac{1}{2}(\mathbf{G} - \mathbf{G}^\top)\}; \;\; \mathbf{A} = \mathbf{Q}\left(\mathbf{R} + \mathrm{diag}\left(\boldsymbol{s}\right)\right), \tag{6}$$

where $\exp(\cdot)$ is the matrix exponential and the learned parameters are $\mathbf{G} \in \mathbb{R}^{d \times d}$, $\mathbf{R} \in \mathbb{R}^{d \times d}$ an upper triangular matrix with zeros on the diagonal, and the vector $\boldsymbol{s} \in \mathbb{R}^d$. $\mathbf{Q}$ is orthogonal by construction as the exponent operates on a skew-symmetric matrix, and $\mathbf{R} + \mathrm{diag}\left(\boldsymbol{s}\right)$ by design is an upper triangular matrix.

Using the parameterizations in Eq. 5 & 6, ensuring that $\mathbf{A}$ is invertible amounts to enforcing that $\det\left(\mathrm{diag}(\mathbf{s})\right) > 0$. We control it by initializing $\mathbf{A}$ to be a block-diagonal rotation matrix and using the following regularization,

$$\mathcal{L}_{vol} = (\prod_{i=1}^d |\boldsymbol{s}_i| - 1)^2. \tag{7}$$

As $\left|\det\left(\mathbf{A}\right)\right| = \prod_i |\boldsymbol{s}_i|$, this regularization essentially attempts to keep the absolute determinant of the transformation close to one, pushing the transformation to be volume preserving. The goal of the regularization is two-fold: (1)

to mitigate numerical instabilities, and (2) to allow individual directions to expand or contract while preserving the overall volume of the activation region. In practice, instead of learning $s$ directly, we learn $\log |s|$ and regularize $(\sum_{i=1}^{d} \log |s_i|)^2$ to be close to zero. This objective shares the same minima as that in Eq. 7, but is more stable.

**Learning the transformations**. We now turn to describe the optimization procedure of LATMiX. Let $f(\cdot)$ denote the full-precision LLM with $N$ transformer layers. Our goal is to learn $N + 1$ sets of parameters of the form $\{\mathbf{L}, \mathbf{U}, s, v\}$ for the LU parametrization and $\{\mathbf{G}, \mathbf{R}, s, v\}$ for the QR parametrization. One set for parameterizing $\mathbf{T}_1$ and $N$ such sets, one for each transformer block, for parameterizing $\mathbf{T}_2$. Denote by $\Omega$ the set of all learnable parameters and by $\tilde{f}_\Omega(\cdot)$ the network corresponding to $f(\cdot)$ with the activations transformed according to $\mathbf{T}_1$ and $\mathbf{T}_2$ and then quantized using MX. In the literature, orthogonal transformations are chosen to uphold the computational invariance theorem (Ashkboos et al., 2024a;c), as they do not modify the output of RMSNorm layers, namely $\frac{x}{||x||_2} = \mathbf{A}^{-1} \frac{\mathbf{A}x}{||\mathbf{A}x||_2}$. In our case, since we allow general invertible matrices, it is no longer true as $\mathbf{A}$ does not have to be orthogonal and we have a bias term $v$. Hence, we relax this restriction by initializing $\mathbf{T}$ to be a rotation transformation and learn $\Omega$ using a distillation loss on a small set of examples (Hinton et al., 2015; Liu et al., 2024; Hu et al., 2025),

$$\mathcal{L}_{dist} = \text{KL}\left(f(x), \tilde{f}_\Omega(x)\right). \tag{8}$$

Here, KL refers to Kullback-Leibler divergence, $f(x)$ is the teacher and $\tilde{f}_\Omega(x)$ is the student. At the start of training, the initialization enforces consistency between the models. During optimization, the loss allows traversing the transformation space to reach a better solution while approximately maintaining consistency between the models. Since common quantization methods (e.g., GPTQ) already rely on an unlabeled calibration set, we reuse a subset of these examples to learn $\Omega$ and evaluate the resulting model on various tasks, including zero-shot tasks. To summarize, the final loss for learning $\Omega$ is:

$$\mathcal{L} = \mathcal{L}_{dist} + \lambda \cdot \mathcal{L}_{vol}, \tag{9}$$

where $\lambda$ is a hyperparameter controlling the effect of the regularization. In our experiments, we found that LATMiX was extremely robust to the selection of $\lambda$. In Appendix E.3, we further compare the loss in Eq. 8 to per-block MSE loss, and theoretically connect it to standard next token prediction cross-entropy loss (Liu et al., 2025). We found that the KL loss is preferred in zero-shot settings over both alternatives, where the transformations are learned by solving a language modeling task and used for other tasks. Importantly, this does not come at the expense of efficiency, as LATMiX training runtime is comparable to that of popular calibration-based approaches (Liu et al., 2025), requiring approximately two hours to train Qwen3-14B on 4 A100 GPUs.

**Weight quantization**. After completing the optimization stage, the transformations $\mathbf{T}_1$ and $\mathbf{T}_2$ are folded into the linear operations of the model. A detailed description of the folding process is provided in Appendix C. A quantization procedure is then applied, such as GPTQ (Frantar et al., 2023), to compensate for weight quantization error.

# 4. Related Work

A prominent approach to reduce the inference cost in LLMs is post-training quantization (PTQ) (Lin et al., 2016; Nagel et al., 2020). In recent years, several studies have proposed methods for quantizing transformer-based LMs, (Shen et al., 2020; Yao et al., 2022; Lee et al., 2023; Frantar et al., 2023), to name a few. Two central aspects of quantization are the numerical representation format of model elements (e.g., FP4, MXFP4), and the mitigation of outliers to reduce errors prior to quantization. In this study, we focus mainly on MX (Rouhani et al., 2023b) as it was shown to be highly effective for language models (Rouhani et al., 2023a).

As outliers are prominent in LLMs, many approaches have been proposed to tackle them. Several attempts used grouping and mixed precision quantization (Bondarenko et al., 2021; Dettmers et al., 2022; 2024; Ashkboos et al., 2024b; Kim et al., 2024; Zhao et al., 2024; Ramachandran et al., 2025), which limits their applicability in cases of strict low-bit precision constraints for all quantities and requires costly operations. Other approaches build on invertible transformations, such as reparameterizing activations and weights through shifting and scaling operations (Wei et al., 2022b; 2023; Shao et al., 2024), scaling the weights based on activation information (Xiao et al., 2023; Liu et al., 2023; Lin et al., 2024b), and multiplying the weights per-layer with random orthogonal (Chee et al., 2023), Hadamard (Tseng et al., 2024), and rotation (Ashkboos et al., 2024c) matrices to spread the magnitude and sensitivity across coordinates. LATMiX generalizes these approaches by allowing *general* affine transformation. Instead of using random rotations to mitigate outliers, several studies proposed different alternatives to construct and learn rotation matrices (Lin et al., 2024a; Liu et al., 2025; Akhondzadeh et al., 2025; Shao et al., 2025a). Lin et al. (2024a) proposed using greedy search combined with permutation of (non-MX) blocks to obtain a balanced distribution of outliers. However, this method fails to account for the MX block structure. Liu et al. (2025) proposed learning rotation matrices on the Stiefel manifold , while Shao et al. (2025a) learns square matrices and retains only the rotation component using QR decomposition. LATMiX instead learns free-form matrices, allowing us to use the full machinery of deep learning optimization (Kingma & Ba, 2015; Loshchilov & Hutter, 2019). The

*Table 1.* Zero-shot average accuracy (Acc.) and average recovery (Rec.) in percentages on seven commonsense and reading-comprehension benchmarks. All methods were evaluated under the same experimental setup. All methods except RTN and QuaRot-RTN were combined with GPTQ quantization scheme. FlatQuant[†] - Using FlatQuant's matrix structure in our experimental setup (further details in Appendix D.2). Within each model, the best method is in **bold** and the second best is underlined.

| Format | Method | Llama-3.2-1B | | Llama-3.2-3B-Instruct | | Llama-3.1-8B-Instruct | | Qwen3-1.7B | | Qwen3-4B | | Qwen3-8B | | Qwen3-14B | | Qwen3-32B | |
|---|---|---|---|---|---|---|---|---|---|---|---|---|---|---|---|---|---|
| | | Acc. | Rec. | Acc. | Rec. | Acc. | Rec. | Acc. | Rec. | Acc. | Rec. | Acc. | Rec. | Acc. | Rec. | Acc. | Rec. |
| | FP16 | 57.53 | 100 | 63.54 | 100 | 70.90 | 100 | 60.04 | 100 | 67.02 | 100 | 70.01 | 100 | 73.29 | 100 | 74.29 | 100 |
| MXFP4 | RTN | 48.78 | 84.58 | 59.97 | 94.06 | 66.77 | 93.59 | 51.39 | 85.30 | 59.76 | 88.93 | 64.72 | 92.21 | 69.80 | 94.88 | 70.91 | 95.66 |
| | QuaRot-RTN | 46.39 | 80.00 | 54.25 | 84.13 | 62.42 | 87.40 | 46.72 | 77.02 | 56.00 | 83.84 | 63.33 | 89.95 | 68.30 | 92.85 | 69.89 | 94.01 |
| | GPTQ | 49.22 | 85.29 | 59.42 | 92.92 | 67.44 | 94.73 | 50.99 | 84.71 | 62.11 | 92.54 | 65.04 | 92.74 | 70.60 | 96.12 | 71.40 | 96.95 |
| | QuaRot | 51.74 | 89.64 | 58.72 | 91.74 | 66.66 | 93.32 | 51.81 | 86.48 | 61.61 | 91.47 | 65.46 | 93.15 | 71.33 | 97.18 | 72.32 | 97.39 |
| | SpinQuant | 51.81 | 90.06 | 59.68 | 93.17 | 68.10 | 95.56 | 52.70 | 87.20 | 60.37 | 90.04 | 66.22 | 94.53 | 70.95 | 96.59 | 70.83 | 95.56 |
| | OSTQuant | 52.57 | 91.22 | 59.36 | 92.64 | 67.02 | 94.04 | 55.48 | 92.35 | 62.42 | 93.10 | 67.28 | 96.05 | 71.74 | 97.67 | 73.25 | 98.82 |
| | FlatQuant[†] | 52.45 | 90.54 | 60.55 | 94.73 | 67.80 | 94.92 | 54.85 | 90.98 | 62.46 | 92.96 | 66.89 | 95.34 | 70.56 | 96.07 | 71.27 | 95.60 |
| | MR-GPTQ | 53.52 | 93.07 | 60.65 | 95.02 | 68.93 | **96.71** | 52.23 | 86.59 | 62.21 | 92.70 | 67.71 | 96.91 | 71.33 | 97.39 | 72.67 | 98.01 |
| | **LATMiX-LU (Ours)** | **54.04** | **93.88** | **62.61** | **97.95** | 68.45 | 96.17 | **57.42** | **95.62** | **64.74** | **96.64** | 67.94 | 97.07 | 71.82 | 97.88 | 73.44 | **99.04** |
| | **LATMiX-QR (Ours)** | 53.79 | 93.42 | 61.61 | 96.59 | 68.44 | 96.21 | 56.68 | 93.74 | 64.08 | 95.64 | 68.64 | **97.97** | 71.36 | 97.17 | 72.19 | 97.01 |
| MXINT4 | RTN | 43.98 | 76.32 | 55.88 | 87.31 | 64.62 | 90.64 | 46.40 | 76.94 | 50.92 | 76.31 | 59.97 | 85.25 | 66.81 | 90.88 | 63.33 | 85.19 |
| | QuaRot-RTN | 44.69 | 77.90 | 50.08 | 78.02 | 59.33 | 83.18 | 42.22 | 69.77 | 53.74 | 79.42 | 59.23 | 84.17 | 65.76 | 88.92 | 68.43 | 91.91 |
| | GPTQ | 43.61 | 75.16 | 55.96 | 86.79 | 63.68 | 89.04 | 46.18 | 76.75 | 55.09 | 82.49 | 61.89 | 88.42 | 67.59 | 91.83 | 70.02 | 94.15 |
| | QuaRot | 49.90 | 86.95 | 57.58 | 90.13 | 65.76 | 92.20 | 49.72 | 82.80 | 58.31 | 86.62 | 63.97 | 91.35 | 70.74 | 96.19 | 71.84 | 96.64 |
| | SpinQuant | 49.54 | 85.81 | 60.02 | 93.83 | 66.65 | 93.59 | 49.53 | 81.55 | 56.29 | 84.14 | 63.68 | 90.43 | 70.11 | 95.39 | 69.97 | 94.15 |
| | OSTQuant | 50.43 | 87.51 | 57.70 | 90.11 | 66.13 | 92.87 | 54.37 | 90.47 | 61.06 | 90.82 | 67.02 | 95.82 | 71.19 | 96.97 | 73.06 | 98.32 |
| | FlatQuant[†] | 50.34 | 87.25 | 58.31 | 90.77 | 66.04 | 92.35 | 53.12 | 88.56 | 61.50 | 91.50 | 65.80 | 93.76 | 69.14 | 94.36 | 70.66 | 95.07 |
| | MR-GPTQ | 51.07 | 88.21 | 60.68 | 95.11 | 68.24 | 95.89 | 51.77 | 86.27 | 61.01 | 91.02 | 66.68 | 95.01 | 70.98 | 96.53 | 72.30 | 97.51 |
| | **LATMiX-LU (Ours)** | 52.24 | 90.34 | 60.77 | 95.15 | 68.44 | **96.20** | 55.43 | 92.24 | 63.63 | **94.89** | 68.10 | **97.16** | 71.79 | **97.96** | **73.84** | **99.49** |
| | **LATMiX-QR (Ours)** | 52.39 | **91.17** | 61.17 | **95.94** | 68.15 | 95.54 | 55.87 | **92.96** | 63.70 | 94.73 | 67.09 | 96.00 | 70.39 | 95.86 | 71.54 | 96.16 |

studies closest to ours are (Ma et al., 2024) and (Sun et al., 2025), which used invertible transformations, although not under MX quantization. In both (Ma et al., 2024) and (Sun et al., 2025), the transformations were restricted to a subclass of all invertible matrices, namely strictly diagonally dominant matrices in the former and Kronecker product of low-rank matrices in the latter. Also, unlike LATMiX, both approaches learn only local matrices per transformer block online using a local loss. As a result, some transformations are limited to scaling transformations only (Ma et al., 2024), or are computationally more costly, as folding is not supported (Sun et al., 2025). In Section 5, we compare to the latter, showing a consistent benefit for LATMiX over it. Several studies have explored outlier reduction under MX quantization. Sharify et al. (2024) adapt GPTQ (Frantar et al., 2023) to the block-wise setting and combine it with SmoothQuant (Xiao et al., 2023) and AWQ (Lin et al., 2024b) for outlier reduction. Lee et al. (2025b) identified that rotation of activations and microscaling lack synergy and proposed a new format using asymmetric scales shared within a quantization group. Likewise, Lee et al. (2025a) proposed to modify the MX format by re-purposing the exponent field to store more mantissa bits. Unlike these studies, we focus on outlier mitigation without modifying the PTQ method or data format. Lastly, (Egiazarian et al., 2025) and (Shao et al., 2025b) recently studied quantization under the MX format. Both studies advocate for using block diagonal rotation matrices, instead of full rotation matrices. Our analysis shows that it can indeed be beneficial; however, different from these studies, we found that adopting and learning full affine transformations that account for the

feature distribution and the MX block structure can mitigate outliers more effectively.

# 5. Experiments

We evaluate LATMiX performance on various language models and tasks to demonstrate its effectiveness. We conducted experiments on Llama3.2 (AI, 2024) (1B/3B) Llama3.1 (Dubey et al., 2024) (8B), and Qwen3 (Yang et al., 2025) (1.7B/4B/8B) models. We focus primarily on small-scale models, as our interest lies in regimes of extreme quantization. In the main text, we present results on 7 zero-shot commonsense reasoning tasks. Appendix E presents additional results and ablation studies, supporting different design choices for LATMiX, including perplexity on the WikiText2 dataset (Merity et al., 2016) in Appendix E.1.

## 5.1. Experimental Settings

The optimization of LATMiX proceeds in two stages. First, invertible affine transformations are learned for $T_1$ and $T_2$ as described in Section 3.2. In the second stage, block-wise GPTQ is applied to the transformed weights similarly to previous studies (e.g., (Egiazarian et al., 2025)). We compare LATMiX against several methods spanning both MX quantization alone and MX combined with different transformation-based approaches for outlier suppression. We first evaluate MX quantization without transformations. These experiments are denoted as **RTN** for plain round-to-nearest quantization, and as **GPTQ** when using GPTQ as the underlying quantization scheme. We then combine

*Table 2.* Transformation and granularity ablation on WikiText2.

| Transformation | Granularity | Llama3.2-1B | Qwen3-1.7B |
|---|---|---|---|
| None | – | 18.80 | 21.29 |
| Random Hadamard | Block | 12.25 | 21.30 |
| | Full | 12.87 | 20.24 |
| Learned Orth. Matrix (Ours) | Block | 13.89 | 18.20 |
| | Full | 13.35 | 17.70 |
| Learned Orth. Matrix + bias (Ours) | Block | 14.43 | 24.04 |
| | Full | 11.99 | 17.28 |
| Learned Inv. Matrix (Ours) | Block | 12.16 | 21.11 |
| | Full | 11.75 | 17.50 |
| **LATMiX-LU (Ours)** | Block | 12.08 | 18.13 |
| | **Full** | **11.64** | **15.11** |

*Table 3.* Qwen3-8B full-precision model perplexity after fusing the learned $\mathbf{T}_1$ and $\mathbf{T}_2$, at multiple training steps.

| Training steps | FP16 | 0 | 1 | 100 | 500 | 1000 |
|---|---|---|---|---|---|---|
| **PPL ($\downarrow$)** | 10.246 | 10.243 | 10.242 | 10.240 | 10.248 | 10.236 |

each method with tensor-wise or block-wise transformations to emulate prior approaches, including QuaRot (Ashkboos et al., 2024c), SpinQuant (Liu et al., 2025), OSTQuant (Hu et al., 2025), FlatQuant (Sun et al., 2025), and MR-GPTQ (Egiazarian et al., 2025), and evaluate them on the same benchmarks. In the main text, we compared against FlatQuant's matrix structure under the same transformation pipeline used for all methods, including LATMiX. This enabled a direct evaluation of our proposed affine transformation constructions based on QR and LU decompositions against alternative approaches, such as Kronecker products of low-rank matrices. A comparison with the original FlatQuant formulation is provided in Appendix D.2, where we observe better performance for LATMiX in that case as well. Transformations are learned using 256 samples from the WikiText2 dataset (Merity et al., 2016). All experiments were conducted using the codebase provided by (Egiazarian et al., 2025) with MX block size set to 32. We follow the implementation of (Liu et al., 2025) and execute all methods under the same experimental setup to ensure fairness in the comparisons. Appendix D provides full implementation details.

## 5.2. Zero-shot Reasoning Tasks Evaluation

We evaluate zero-shot accuracy on a suite of commonsense and reading-comprehension benchmarks: ARC (Easy and Challenge) (Clark et al., 2018), HellaSwag (Zellers et al., 2019), WinoGrande (Sakaguchi et al., 2021), PIQA (Bisk et al., 2020), BoolQ (Clark et al., 2019), and OpenBookQA (OBQA) (Mihaylov et al., 2018). All results are obtained using the LM Evaluation Harness (Gao et al., 2024) with default task settings. Performance is summarized in Table 1, where we report, for each quantized model, the mean accuracy and mean recovery relative to the floating-point baseline averaged across tasks. Full per-benchmark results are provided in Appendix F. From the table, consistent with prior work (Egiazarian et al., 2025; Shao et al., 2025b), combining RTN with QuaRot leads to performance degradation. In addition, while learning the transformations as in SpinQuant and FlatQuant or using block-wise transfor-

mations improves quantization performance, these methods remain limited and do not fully exploit the potential of transformation-based outlier suppression under MX quantization. In contrast, LATMiX achieves the best overall performance, indicating that explicitly accounting for both the MX quantization structure and the activation distribution via invertible affine transformations is more effective. Appendix E.1 further reports perplexity results on WikiText2, showing that LATMiX outperforms baseline methods on that benchmark as well. Appendix E.6 compares LATMiX to baseline approaches under NVFP quantization. Here as well, LATMiX outperforms baseline methods in most cases, at times almost reaching FP16 performance.

Furthermore, examining both LU and QR variants of LATMiX, we observe a small advantage for the LU over QR parameterization. We attribute this to the QR optimization process, as it requires backpropagation through a matrix exponential. We view this as a promising direction for future research, as in principle the QR variant should obtain better performance throughout. In Appendix E.4 we show that both LU and QR parametrization of $\mathbf{T}_1$ are well-conditioned, indicating that they can be inverted stably.

## 5.3. Ablation & Analysis

**Transformation Type Analysis**. To study the effect of different components in LATMiX, we evaluate the perplexity on WikiText2 of MXFP4-quantized models under different types of transformations for both $\mathbf{T}_1$ and $\mathbf{T}_2$. We consider two baselines: (1) None - without transformations; and (2) Random Hadamard matrices. We also evaluate three variants of our approach: (3) Orthogonal transformations (with and without learned bias), learning only $\mathbf{Q}$ in the QR parametrization; (4) Invertible transformations, learning $\mathbf{A}$ only using LU decomposition; and (5) the LU variant of LATMiX. Each method is applied either at the tensor level (Full) or in a block-wise manner (Block). Results on WikiText2 perplexity are shown in Table 2. From the table, learning invertible transformations is preferred over both learnable orthogonal transformations and random rotations, but underperforms learning affine transformations. LATMiX applied at the tensor level consistently achieves the best performance across models. These findings indicate that learning both a general invertible linear transform and a bias term at full granularity is more effective than fixed rotations or block-wise alternatives for mitigating activation outliers. Moreover, we observe that block-wise Hadamard rotation outperforms full Hadamard rotation on Llama3.2-

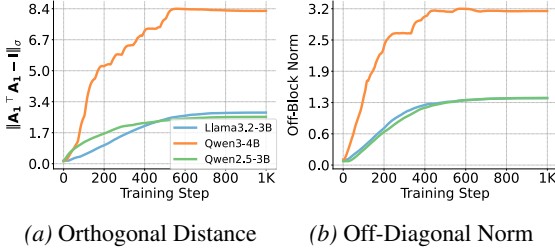

*(a)* Orthogonal Distance     *(b)* Off-Diagonal Norm

*Figure 3.* LATMiX learns a transformation that spreads the energy across the tensor.

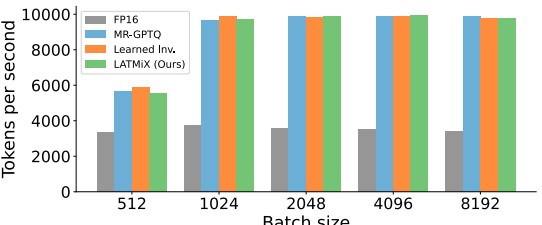

*Figure 4.* Throughput of different approaches.

1B, while this trend does not hold for Qwen3-1.7B. This emphasizes that both the quantization structure and the feature distribution should be taken into account.

**Learned Transformation Analysis**. To better understand the behavior of our learned transformation, we report two metrics in Fig. 3. Fig. 3a shows the orthogonality deviation of the learned transform matrix $\mathbf{A_1}$ by measuring the distance from an orthogonal transformation using the spectral norm. Fig. 3b measures the spectral norm of the off-block-diagonal components of the learned transformation, achieved by assigning zeros to the block-diagonal elements. An increase in this metric suggests that the learned matrix departs from purely block-diagonal behavior, enabling interactions across blocks. From Fig. 3a, the orthogonality deviation increases rapidly during the early stages of training from the initialized orthogonal matrix. This indicates that the optimization quickly moves away from the orthogonal constraint. After this initial phase, the deviation stabilizes, suggesting convergence to a non-orthogonal but more effective solution for quantization. Furthermore, Fig. 3b shows that the optimization is indeed adapted to the MX quantization structure, moving beyond the redistribution of outliers intra-block and allowing the transfer of energy between blocks. Additional analysis of the learned transformation deviation from the orthogonal initial point and the effect of each transformation can be found in Appendix E.2 and Appendix E.5.6, respectively.

**Computational Invariance**. A distinguishing feature of LATMiX relative to prior work is that it relaxes the requirement for strict computational invariance (Ashkboos et al., 2024a). Here, we show that LATMiX, using the proposed distillation loss, keeps the transformed quantized model approximately consistent with the float model by showing that the learned transformations neither alter the network behavior nor lead to overfitting on the calibration dataset. To assess the method's sensitivity, we first note that zero-shot accuracy consistently improves over existing baselines and, in some cases, remains close to full-precision performance, as shown in Table 1. Since calibration uses only the WikiText2 training set, these results suggest that harmful overfitting does not occur in practice. In addition, Table 3

reports the FP16 performance when applying the learned transformations under MXFP4. Specifically, we evaluate the Qwen3-8B FP16 model after fusing the learned $\mathbf{T}_1$ and $\mathbf{T}_2$, without any quantization, at multiple training steps. Compared to the FP16 baseline, the transformed models show negligible changes in WikiText2 test perplexity, indicating that the transformations preserve network behavior. Appendix E.5 further presents additional ablation studies showing that LATMiX performance remains robust to the choice of calibration set and its number of samples.

**Computational Cost**. Here we benchmark LATMiX inference against competing approaches on an NVIDIA RTX 6000 PRO using optimized kernels from (Egiazarian et al., 2025) and vLLM (Kwon et al., 2023). In particular, we compare LATMiX to: (a) BF16, (b) MR-GPTQ, and (c) Learned Inv transformation (LATMiX without the bias term). Figure 4 reports throughput in tokens per second on a range of batch sizes. We observe comparable throughput across all quantized methods, suggesting that, in practice, LATMiX incurs at most negligible inference overhead.

## 6. Conclusion

In this work, we analyze the quantization error of the MX format and demonstrate the importance of incorporating both structural knowledge and feature distributions. Building on this analysis, we introduce LATMiX, a method that enables affine transformations to be applied without runtime or memory overhead, while remaining comparable to leading approaches in this area. In our experiments, we showcase the effectiveness of LATMiX across seven benchmarks, as well as perplexity evaluation on WikiText2, achieving up to a 4% improvement in average accuracy recovery. Overall, our results indicate that learning affine transformations for MX formats in LLMs can yield substantial gains. While we did not evaluate LATMiX on non–MX data types in this work, its performance in such settings remains an interesting direction for future study. One limitation of LATMiX, shared with other optimization-based transformation methods, is the need for training; nevertheless, empirical results clearly demonstrate its benefits.

## Impact Statement

This paper presents work whose goal is to advance the field of Machine Learning. There are many potential societal consequences of our work, none of which we feel must be specifically highlighted here.

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

## A. MX Quantization Error Upper Bound (Theorem 3.3)

We split the proof of Theorem 3.3 into two parts. In the first part, we derive an upper bound without making any assumptions about the distribution of the feature map (i.e., the random vector $\boldsymbol{x}$). In the second part, we compute the expectation of the maximum absolute value of a $\psi_\alpha$ distribution vector and assume that $\boldsymbol{x}$ follows a $\psi_\alpha$ distribution to gain additional insight. We then combine these two parts to complete the proof.

The main assumption required for the proof is that the density of $\boldsymbol{x}$ is bounded. More formally,

**Assumption A.1.** Assume that $\boldsymbol{x}$ is a continuous random vector and that there exists a finite constant $C < \infty$ such that its probability density function $f(\boldsymbol{x})$ satisfies $0 \leq f(\boldsymbol{x}) < C \quad \forall \boldsymbol{x} \in \mathbb{R}^d$.

**Lemma A.2.** *Assume that* $\mathbf{T}$ *is an affine transformation and* $Q$ *is MX quantization according to* (1). *Then,*

$$\mathcal{E}(\mathbf{T}) \lesssim \frac{\|\mathbf{A}^{-1}\|_\sigma^2}{N_B} \sum_{i=1}^{N_B} \mathbb{E}\left[\left(\max_{j\in\mathcal{I}_i}\left|\left[\mathbf{T}(\boldsymbol{x})\right]_j\right|\right)^2\right]. \tag{10}$$

*Proof.* We begin with the definition of quantization error in (2)

$$\mathcal{E}(\mathbf{T}) \triangleq \frac{1}{d}\mathbb{E}\left[\left\|\boldsymbol{x}-\mathbf{T}^{-1}\left(Q\left(\mathbf{T}(\boldsymbol{x})\right)\right)\right\|_2^2\right] = \frac{1}{d}\mathbb{E}\left[\left\|\mathbf{A}^{-1}\left(\mathbf{T}(\boldsymbol{x})-Q\left(\mathbf{T}(\boldsymbol{x})\right)\right)\right\|_2^2\right] \leq \frac{\|\mathbf{A}^{-1}\|_\sigma^2}{d}\mathbb{E}\left[\left\|\mathbf{T}(\boldsymbol{x})-Q\left(\mathbf{T}(\boldsymbol{x})\right)\right\|_2^2\right]$$

$$= \frac{\|\mathbf{A}^{-1}\|_\sigma^2}{N_B\cdot B}\mathbb{E}\left[\left\|\boldsymbol{y}-Q(\boldsymbol{y})\right\|_2^2\right] = \frac{\|\mathbf{A}^{-1}\|_\sigma^2}{N_B\cdot B}\sum_{i=1}^{N_B}\mathbb{E}\left[\left\|\boldsymbol{y}^{(i)}-Q\left(\boldsymbol{y}^{(i)}\right)\right\|_2^2\right], \tag{11}$$

where $\boldsymbol{y}=\mathbf{T}(\boldsymbol{x})$ is introduced for notational convenience and $\boldsymbol{y}^{(i)}=\left[\boldsymbol{y}_{i\cdot B} \quad \cdots \quad \boldsymbol{y}_{(i+1)\cdot B-1}\right]$ represents the vectorized features of the $i^{th}$ MX-Block. In Eq. (2), the first equality follows from the invertibility of $\mathbf{A}$ and the definition of the transformation:

$$\left\|\boldsymbol{x}-\mathbf{T}^{-1}\left(Q\left(\mathbf{T}(\boldsymbol{x})\right)\right)\right\|_2^2 = \left\|\boldsymbol{x}-\mathbf{A}^{-1}Q\left(\mathbf{T}(\boldsymbol{x})\right)+\mathbf{A}^{-1}\boldsymbol{v}\right\|_2^2 = \left\|\mathbf{A}^{-1}\left(\mathbf{A}\boldsymbol{x}+\boldsymbol{v}-Q\left(\mathbf{T}(\boldsymbol{x})\right)\right)\right\|_2^2$$

$$= \left\|\mathbf{A}^{-1}\left(\mathbf{T}(\boldsymbol{x})-Q\left(\mathbf{T}(\boldsymbol{x})\right)\right)\right\|_2^2.$$

Next, we examine the error in each block and MX Quantization definition in (1):

$$\mathbb{E}\left[s_i^2\left\|\frac{\boldsymbol{y}^{(i)}}{s_i}-Q_e\left(\frac{\boldsymbol{y}^{(i)}}{s_i}\right)\right\|_2^2\right] = \mathbb{E}\left[s_i^2\sum_{j=1}^B\left(\frac{\boldsymbol{y}_j^{(i)}}{s_i}-Q_e\left(\frac{\boldsymbol{y}_j^{(i)}}{s_i}\right)\right)^2\right] = \mathbb{E}\left[s_i^2\sum_{j=1}^B\mathbb{E}\left[\left(\frac{\boldsymbol{y}_j^{(i)}}{s_i}-Q_e\left(\frac{\boldsymbol{y}_j^{(i)}}{s_i}\right)\right)^2\Big|s\right]\right]. \tag{12}$$

In Eq. (12), the first step splits the error by block, and the second applies the law of total expectation. We now derive the error for each element within a block.

$$\mathbb{E}\left[\left(\frac{\boldsymbol{y}_j^{(i)}}{s_i}-Q_e\left(\frac{\boldsymbol{y}_j^{(i)}}{s_i}\right)\right)^2\Big|s\right] = \int_{\boldsymbol{y}_j^{(i)}}\left(\frac{\boldsymbol{y}_j^{(i)}}{s_i}-Q_e\left(\frac{\boldsymbol{y}_j^{(i)}}{s_i}\right)\right)^2 f_{\boldsymbol{y}_j^{(i)}|s_i}\left(\boldsymbol{y}_j^{(i)}|s_i\right)d\boldsymbol{y}_j^{(i)}. \tag{13}$$

Apply a change in variable $z=\frac{\boldsymbol{y}_j^{(i)}}{s_i}$, $d\boldsymbol{y}_j^{(i)}=s_i dz$ and combining with the definition of MX in (1) we assume that the domain of $z$ is in the range of $Q_e$, results in

$$s_i\int_z\left(z-Q_e(z)\right)^2 f_{\boldsymbol{y}_j^{(i)}|s_i}\left(z\cdot s_i|s_i\right)dz = s_i\sum_{k=1}^{|\mathcal{Q}|}\int_{l_k}^{u_k}\left(z-q_k\right)^2 f_{\boldsymbol{y}_j^{(i)}|s_i}\left(z\cdot s_i|s_i\right)dz$$

$$= \sum_{k=1}^{|\mathcal{Q}|}\int_{l_k}^{u_k}\left(z-q_k\right)^2 f_{z|s_i}\left(z|s_i\right)dz \leq C_Q \triangleq \sum_{k=1}^{|\mathcal{Q}|}\int_{l_k}^{u_k}\left(z-q_k\right)^2 dz. \tag{14}$$

Where, $q_k$ is the quantized value in the interval between $l_k$ to $u_k$ and in the last step in Eq. (14) we upper bound the probability over the interval between $u_k$ to $l_k$ as we assume that the probability density can be bounded by some constant. Combining Eq.(14), (13), (12) and (11) results in:

$$\mathcal{E}\left(\mathbf{T}\right) \leq \left\|\mathbf{A}^{-1}\right\|_\sigma^2 C_Q \frac{1}{N_B} \sum_{i=1}^{N_B} \mathbb{E}\left[s_i^2\right] = \left\|\mathbf{A}^{-1}\right\|_\sigma^2 C_Q \frac{1}{N_B} \sum_{i=1}^{N_B} \mathbb{E}\left[2^{2\left\lfloor \log_2\left(\max_{j\in\mathcal{I}_i}|\boldsymbol{y}_j|\right)\right\rfloor - 2r_{max}}\right]$$

$$\leq \left\|\mathbf{A}^{-1}\right\|_\sigma^2 C_Q \frac{2^{-2r_{max}}}{N_B} \sum_{i=1}^{N_B} \mathbb{E}\left[\left(\max_{j\in\mathcal{I}_i}|\boldsymbol{y}_j|\right)^2\right]. \tag{15}$$

In (15), the first equality follows from the definition of MX quantization in (1), and the second inequality follows from the definition of the floor function. □

Lemma A.2 establishes the first part of Theorem 3.3, namely inequality (3). To prove the second part of Theorem 3.3, we now present and prove an upper bound on the expected maximum of a $\psi_\alpha$ distributions.

The $\psi_\alpha$ distributions form a class of probability distributions whose tails decay at least as fast as $\exp(-c|x|^\alpha)$. We begin by stating a tail bound for $\psi_\alpha$ distributions (Vershynin, 2026):

**Theorem A.3** ($\psi_\alpha$ distributions tail). *Let $\boldsymbol{Z}$ be distributed according to the $\psi_\alpha$ distribution and suppose that $\|\boldsymbol{Z}\|_{\psi_\alpha} \leq K$. Then:*

$$\mathbb{P}\left(|\boldsymbol{Z}| > t\right) \leq 2\exp\left(-\left(\frac{t}{K}\right)^\alpha\right), \tag{16}$$

*where* $\|\boldsymbol{Z}\|_{\psi_\alpha} \triangleq \inf\left\{c > 0 : \mathbb{E}\left[\exp\left(\frac{|\boldsymbol{Z}|^\alpha}{c^\alpha}\right)\right] \leq 2\right\}$ *is the Orlicz norm of the random variable $\boldsymbol{Z}$*

**Lemma A.4.** *Let $\boldsymbol{y} \in \mathbb{R}^B$ be a random vector with mean $\boldsymbol{\mu}$, $\|X\|_{\psi_\alpha}$ is Orlicz norm of random variable $X$. Assume that $\boldsymbol{y}$ is a $\psi_\alpha$ random vector. Then,*

$$\mathbb{E}\left[\left(\max_j|\boldsymbol{y}_j|\right)^p\right]^{1/p} \leq \max_j\left|[\boldsymbol{\mu}]_j\right| + \max_i\left\|[\boldsymbol{y} - \boldsymbol{\mu}]_i\right\|_{\psi_\alpha}\left(\left(\log(2B)\right)^{p/\alpha} + \frac{2B\cdot p}{\alpha}\cdot\Gamma\left(p/\alpha, \log(2B)\right)\right)^{1/p} \tag{17}$$

*Proof.* We start by centering $\boldsymbol{y}$ and placing an upper bound on its absolute maximum.

$$\max_j|\boldsymbol{y}_j| = \max_j\left|[\boldsymbol{y} - \boldsymbol{\mu} + \boldsymbol{\mu}]_j\right| \leq \max_j\left|[\boldsymbol{y} - \boldsymbol{\mu}]_j\right| + \max_j|\boldsymbol{\mu}_j| \tag{18}$$

Next, by definition of the $L_p$ norm of a random variable, we obtain:

$$\mathbb{E}\left[\left(\max_j|\boldsymbol{y}_j|\right)^p\right]^{1/p} = \left\|\max_j|\boldsymbol{y}_j|\right\|_{L_p} = \left\|\max_j\left|[\boldsymbol{y} - \boldsymbol{\mu}]_j\right| + \max_j|\boldsymbol{\mu}_j|\right\|_{L_p} \leq \max_j|\boldsymbol{\mu}_j| + \left\|\max_j\left|[\boldsymbol{y} - \boldsymbol{\mu}]_j\right|\right\|_{L_p}. \tag{19}$$

In Eq. (19), the first step follows from the definition of the random-variable $L_p$ norm (see (Vershynin, 2026), Chapter 1.3); the second uses the triangle inequality, and the last uses that $\boldsymbol{\mu}$ is deterministic (i.e., not a random variable). For notational convenience, let us denote $z = \max_j\left|[\boldsymbol{y} - \boldsymbol{\mu}]_j\right|$. We then only need to evaluate the second term in (19):

$$\left\|\max_j\left|[\boldsymbol{y} - \boldsymbol{\mu}]_j\right|\right\|_{L_p} = \mathbb{E}\left[z^p\right]^{1/p}. \tag{20}$$

Next, to derive and compute $\mathbb{E}\left[z^p\right]$, we assume that each component $[\boldsymbol{y} - \boldsymbol{\mu}]_i$ follows a $\psi_\alpha$ distribution, and that the largest Orlicz norm among all these variables is bounded by

$$K \triangleq \max_i\left\|[\boldsymbol{y} - \boldsymbol{\mu}]_i\right\|_{\psi_\alpha}.$$

Using this and the tail bound for $\psi_\alpha$ variables, their tail probabilities decay as $\exp\left(-c\frac{t^\alpha}{K^\alpha}\right)$ (Vershynin, 2026). Observe that the $\psi_\alpha$ class generalizes sub-Gaussian and sub-exponential distributions: setting $\alpha = 2$ yields the sub-Gaussian case, while $\alpha = 1$ recovers the sub-exponential case. To compute $\mathbb{E}\left[z^p\right]$ we first, split the integration into two parts as follows:

$$\mathbb{E}\left[z^p\right] = \int_0^\infty pt^{p-1}\mathbb{P}\left(z \geq t\right)dt = \int_0^{t_0} pt^{p-1}\mathbb{P}\left(z \geq t\right)dt + \int_{t_0}^\infty pt^{p-1}\mathbb{P}\left(z \geq t\right)dt$$

$$\leq \int_0^{t_0} pt^{p-1}dt + \int_{t_0}^\infty pt^{p-1}\mathbb{P}\left(z \geq t\right)dt \qquad (21)$$

By selection $t_0 = K\left(\log\left(2B\right)\right)^{1/\alpha}$ we have:

$$\int_0^{t_0} pt^{p-1}dt = t_0^p = K^p\left(\log\left(2B\right)\right)^{p/\alpha}. \qquad (22)$$

As a second step we compute the tail bound of $z$.

$$\mathbb{P}\left(z \geq t\right) = \mathbb{P}\left(\bigcup_i \left|\left[\boldsymbol{y} - \boldsymbol{\mu}_y\right]_i\right| \geq t\right) \leq \sum_i \mathbb{P}\left(\left|\left[\boldsymbol{y} - \boldsymbol{\mu}_y\right]_i\right| \geq t\right) = 2 \cdot B \cdot \exp\left(-\frac{t^\alpha}{K^\alpha}\right), \qquad (23)$$

In Eq. (23), the first equality is valid by definition, the second follows from the union bound, and in the last step we use the fact that $\left[\boldsymbol{y} - \boldsymbol{\mu}_y\right]_i$ is $\psi_\alpha$ distribution.

$$\int_{t_0}^\infty pt^{p-1}\mathbb{P}\left(z \geq t\right)dt \leq 2Bp\int_{t_0}^\infty t^{p-1}\exp\left(-\frac{t^\alpha}{K^\alpha}\right)dt$$

$$= \frac{2Bp}{\alpha}K^p\int_{\log(2B)}^\infty u^{p/\alpha-1}\exp\left(-u\right)du$$

$$= \frac{2Bp}{\alpha}K^p\Gamma\left(p/\alpha, \log\left(2B\right)\right) \qquad (24)$$

In Eq. (24), the first inequality follows from Eq. (23), the second step applies the change of variables $u = t^\alpha/K^\alpha$, which implies $t = Ku^{1/\alpha}$ and $dt = \frac{K}{\alpha}u^{1/\alpha-1}du$. The third step uses the definition of the incomplete Gamma function. Combining Eq. (22) and Eq. (24) into Eq. (21) results in:

$$\left\|\max_j\left|\left[\boldsymbol{y} - \boldsymbol{\mu}\right]_j\right|\right\|_{L_p} = \mathbb{E}\left[z^p\right]^{1/p} \leq K\left(\left(\log\left(2B\right)\right)^{p/\alpha} + \frac{2B \cdot p}{\alpha} \cdot \Gamma\left(p/\alpha, \log\left(2B\right)\right)\right)^{1/p}. \qquad (25)$$

Finally, combining Eq. (25) with Eq. (19) yields the desired results. $\qquad\square$

Defining $\boldsymbol{y} = \mathbf{T}\left(\boldsymbol{x}\right)$, it follows that $\boldsymbol{\mu_y} = \mathbb{E}\left[\boldsymbol{y}\right] = \mathbf{T}\left(\mathbb{E}\left[\boldsymbol{x}\right]\right) = \mathbf{T}\left(\boldsymbol{\mu}\right)$. Applying Lemma A.4 with $p = 2$, we obtain

$$M_i \triangleq \mathbb{E}\left[\left(\max_{j \in \mathcal{I}_i}\left|\left[\mathbf{T}\left(\boldsymbol{x}\right)\right]_j\right|\right)^2\right]$$

$$\leq \left(\mathbf{T}_{max}(\boldsymbol{\mu}) + \max_{j \in \mathcal{I}_i}\left\|\left[\hat{\mathbf{T}}\left(\boldsymbol{x}\right)\right]_j\right\|_{\psi_\alpha}\left(\frac{4B}{\alpha} \cdot \Gamma\left(2/\alpha, \log\left(2B\right)\right) + \left(\log\left(2B\right)\right)^{2/\alpha}\right)^{1/2}\right)^2,$$

where $\mathbf{T}_{max}(\boldsymbol{x}) \triangleq \max_{j \in \mathcal{I}_i}\left|\left[\mathbf{T}\left(\boldsymbol{x}\right)\right]_j\right|$, and $\left[\hat{\mathbf{T}}\left(\boldsymbol{x}\right)\right]_j \triangleq \left[\mathbf{T}\left(\boldsymbol{x}\right) - \mathbf{T}\left(\boldsymbol{\mu}\right)\right]_j$.

## B. Affine Transformation inside MHA

Here, we show that an affine transformation can be incorporated into MHA in the general case, and then we detail the corresponding folding operation. Concretely, let $\mathbf{X} \in \mathbb{R}^{N_T \times d}$ denote the input to the MHA, $\mathbf{P}_h \in \mathbb{R}^{N_T \times N_T}$ the attention

matrix of the $h^{\text{th}}$ head, $\mathbf{W}_V^{(h)}$ the value projection matrix of the $h^{\text{th}}$ head, $\mathbf{W}_o$ the output projection matrix, and $N_T$ is the number of tokens. Since the transformation is applied independently to each token, we define the transformation over multiple tokens as follows:

$$\mathbf{T}_2\left(\mathbf{X}\right) = \mathbf{X}\mathbf{A}_2 + \mathbf{V}_2 \tag{26}$$

and its inverse

$$\mathbf{T}_2^{-1}\left(\mathbf{X}\right) = \mathbf{X}\mathbf{A}_2^{-1} - \mathbf{V}_2\mathbf{A}_2^{-1} \tag{27}$$

where $\mathbf{V}_2 \in \mathbb{R}^{N_T \times d}$ denotes the shift matrix, in which the shift vector is replicated across all tokens. Now, we apply it to the multi-head attention block:

$$\mathbf{Y} = \text{Cat}\left(\left[\mathbf{P}_1\mathbf{T}_2\left(\mathbf{X}\mathbf{W}_V^{(1)}\right), \quad \ldots \quad, \mathbf{P}_H\mathbf{T}_2\left(\mathbf{X}\mathbf{W}_V^{(H)}\right)\right]\right) \tag{28}$$

Next, we apply the inverse transformation to $\mathbf{Y}$ and then perform the output projection, yielding $\mathbf{G}^{-1}\left(\mathbf{Y}\right)\mathbf{W}_O$. Because this transformation operates independently on each token, we can, without loss of generality, analyze a single head:

$$\mathbf{T}_2^{-1}\left(\mathbf{P}_1\mathbf{T}_2\left(\mathbf{X}\mathbf{W}_V^{(1)}\right)\right) = \mathbf{P}_1\mathbf{T}_2\left(\mathbf{X}\mathbf{W}_V^{(1)}\right)\mathbf{A}_2^{-1} - \mathbf{V}_2\mathbf{A}_2^{-1} = \mathbf{P}_1\mathbf{X}\mathbf{W}_V^{(1)}\mathbf{A}_2\mathbf{A}_2^{-1} + \mathbf{P}_1\mathbf{V}_2\mathbf{A}_2^{-1} - \mathbf{V}_2\mathbf{A}_2^{-1}$$
$$= \mathbf{P}_1\mathbf{X}\mathbf{W}_V^{(1)} \tag{29}$$

In the final step of (29), we exploit that $\mathbf{P}$ arises from a softmax and that $\mathbf{V}_2$ is replicated across tokens, which implies: $\mathbf{P}_1\mathbf{V}_2\mathbf{A}_2^{-1} = \mathbf{V}_2\mathbf{A}_2^{-1}$. In this way, we can add an affine transformation to the MHA block.

## C. Folding & Transformation Structure

We adopt the same transformation locations as in (Liu et al., 2025) and adapt them to the micro-scaling setup of (Egiazarian et al., 2025). First, we fold the RMS-Norm parameter into the following linear layer as in (Ashkboos et al., 2024c). Concretely, we apply three transformations at the same places as in (Egiazarian et al., 2025): (i) $\mathbf{T}_1$, a transformation applied to the residual path of the model; (ii) $\mathbf{T}_2$, a transformation applied to the value vectors in multi-head attention; and (iii) $\mathbf{T}_3$ (shown in Fig. 5c), an *online* block Hadamard transformation as in (Egiazarian et al., 2025), with its inverse offline transformation folded into the weights. Now, we describe the folding of the transformations $\mathbf{T}_1$ and $\mathbf{T}_2$ such that those transformations will add zero cost to the inference runtime and model size as they are folded into the existing operation. As described in the main text, we derived a way to learn an affine transformation in an LLM so that it can be absorbed into the adjacent linear operation. This eliminates the need to explicitly handle the RMSNorm (e.g., using orthogonal transformation), since it is already taken into account during the optimization process.

### C.1. Folding of $\mathbf{T}_1$

To fold $\mathbf{T}_1$ we need to take into account the shift $\boldsymbol{v}_1$ and the matrix $\mathbf{A}_1$ due to the residual structure of LLM (see Fig. 5), we only add the shift vector $\boldsymbol{v}_1$ once after the embedding layer and remove every block using the inverse transformation. The matrix $\mathbf{A}_1$ is applied to the output of each block, and we define this transformation as $\widetilde{\mathbf{T}}_1\left(\boldsymbol{x}\right) = \mathbf{A}_1\boldsymbol{x}$. Next, we describe how to fold the $\mathbf{T}_1$:

- **Folding $\mathbf{T}_1^{-1}$:** $\mathbf{T}_1^{-1}$ is folded into the K, Q, and V linear mappings within the multi-head attention block. Denote the linear operator for the K, Q, and V features by $f_\Gamma\left(\boldsymbol{x}\right) = \mathbf{W}_\Gamma\boldsymbol{x} + \boldsymbol{b}_\Gamma$, where $\Gamma \in \{K, Q, V\}$. After applying this transformation, we introduce $\widetilde{f}_X$ to represent the resulting folded operation:

$$\widetilde{f}_\Gamma\left(\boldsymbol{x}\right) \triangleq f_\Gamma\left(\mathbf{T}_1^{-1}\left(\boldsymbol{x}\right)\right) = \mathbf{W}_\Gamma\mathbf{T}_1^{-1}\left(\boldsymbol{x}\right) + \boldsymbol{b}_\Gamma = \mathbf{W}_\Gamma\left(\mathbf{A}_1^{-1}\boldsymbol{x} - \mathbf{A}_1^{-1}\boldsymbol{v}_1\right) + \boldsymbol{b}_\Gamma$$
$$= \underbrace{\mathbf{W}_\Gamma\mathbf{A}_1^{-1}}_{=\widetilde{\mathbf{W}}_\Gamma}\boldsymbol{x} + \underbrace{\boldsymbol{b}_\Gamma - \mathbf{W}_\Gamma\mathbf{A}_1^{-1}\boldsymbol{v}_1}_{=\widetilde{\boldsymbol{b}}_\Gamma}. \tag{30}$$

From (30), it follows that $\widetilde{f}_\Gamma$ can be represented as a linear operator with a weight matrix $\widetilde{\mathbf{W}}_\Gamma$ and a bias vector $\widetilde{\boldsymbol{b}}_\Gamma$.

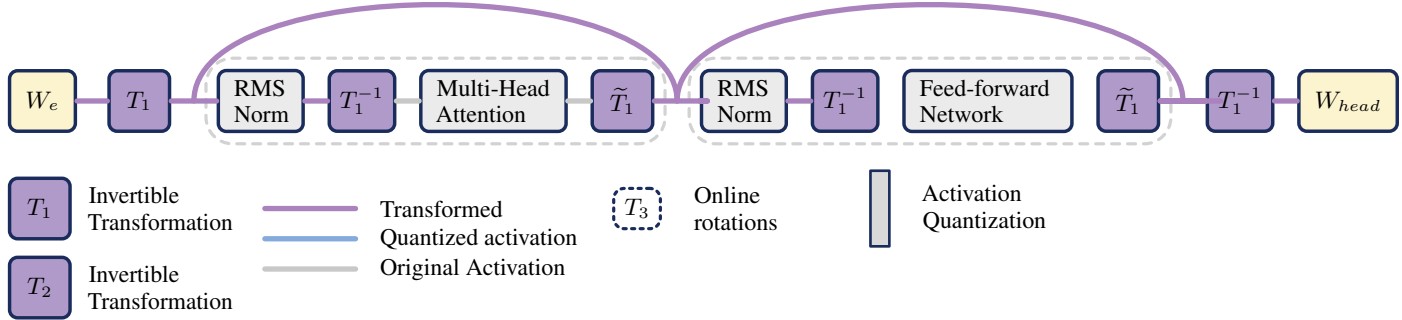

*(a)* Location of $\mathbf{T}_1$ transformation of LLM that enables the folding of general invertible transformation.

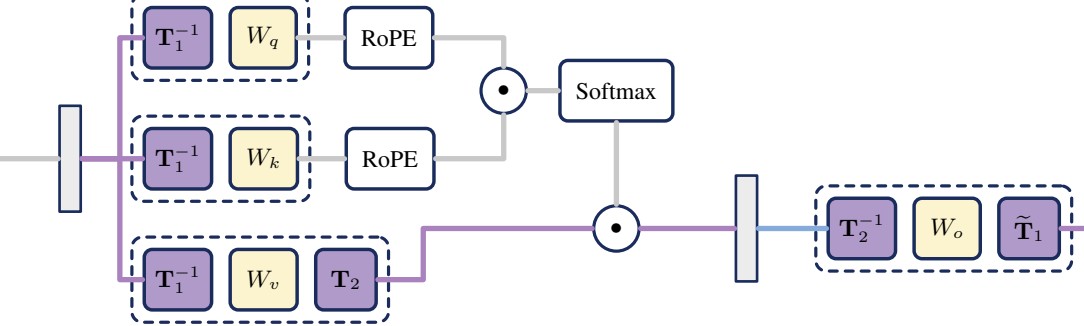

*(b)* Location of $\mathbf{T}_1$ transformation of LLM that enables the folding of general invertible transformation.

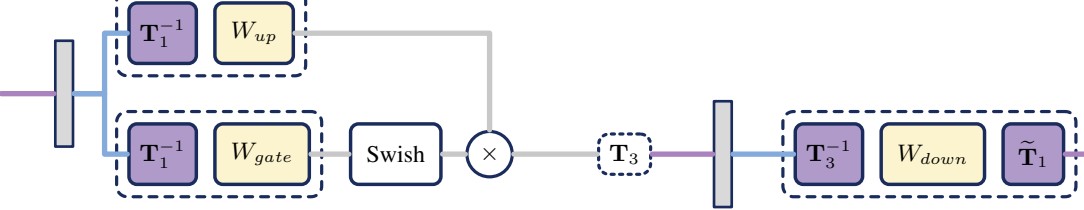

*(c)* Location of $\mathbf{T}_1$ and $\mathbf{T}_3$ transformation of LLM with inside view of Feed-Forward Block.

*Figure 5.* Location of all transformations on a regular LLM with marking of folding operations.

- **Folding $\widetilde{\mathbf{T}}_1$**: The transformation $\widetilde{\mathbf{T}}_1$ can be folded into the linear output operation of the Feed-Forward block and the projection set in the MHA. Specifically, let $f_O(\boldsymbol{x}) = \mathbf{W}_O \boldsymbol{x} + \boldsymbol{b}_O$ be the linear output operation of the FFN and MHA. Applying the transformation to the output of the linear operation yields the following.

$$\widetilde{f}_O(\boldsymbol{x}) = \widetilde{\mathbf{T}}_1\left(f_O(\boldsymbol{x})\right) = \mathbf{A}_1 f_O(\boldsymbol{x}) = \underbrace{\mathbf{A}_1 \mathbf{W}_O}_{=\widetilde{\mathbf{W}}_O} \boldsymbol{x} + \underbrace{\mathbf{A}_1 \boldsymbol{b}_O}_{=\widetilde{\boldsymbol{b}}_O}. \tag{31}$$

From (31), it follows that $\widetilde{f}_O$ can be represented as a linear operator with a weight matrix $\widetilde{\mathbf{W}}_O$ and a bias vector $\widetilde{\boldsymbol{b}}_O$.

- **Folding $\mathbf{T}_1$**: $\mathbf{T}_1$ is applied only to the embedding matrix. Specifically, let $\mathbf{W}_e$ denote the embedding matrix and $\boldsymbol{w}_j$ the $j^{\text{th}}$ embedding vector. We apply the transformation to the feature after the embedding step, which is equivalent to applying the transformation to each embedding vector individually, resulting in:

$$\widetilde{\boldsymbol{w}}_j = \mathbf{T}_1\left(\boldsymbol{w}_j\right) = \mathbf{A}_1 \boldsymbol{w}_j + \boldsymbol{v}_1. \tag{32}$$

This means that $\widetilde{\boldsymbol{w}}_j$ represents the folded embedding vectors.

## C.2. Folding of $\mathbf{T}_2$

After folding the transformation $\mathbf{T}_1$, we describe here the folding of $\mathbf{T}_2$ into the value projection and output projection of the multi-head attention block. Note that there is a separate transformation $\mathbf{T}_2$ for each multi-head attention Block.

Now we fold $\mathbf{T}_2$ into the linear operation $V$ and apply the inverse operation in the MHA output projection.

- **Fold $\mathbf{T}_2$ into values linear projection** Since $\mathbf{T}_1^{-1}$ has already been absorbed into the linear projection of Values as shown in (30), we now add to it $\mathbf{T}_2$, which we denote by $\widetilde{\widetilde{f}}(\boldsymbol{x})$ and compute as follows:

$$\widetilde{\widetilde{f}}(\boldsymbol{x}) \triangleq \mathbf{T}_2\left(\widetilde{f}_V(\boldsymbol{x})\right) = \mathbf{A}_2\widetilde{f}_V(\boldsymbol{x}) + \boldsymbol{v}_2 = \underbrace{\mathbf{A}_2\mathbf{W}_V\mathbf{A}_1^{-1}}_{=\widetilde{\widetilde{\mathbf{W}}}_V}\boldsymbol{x} + \underbrace{\mathbf{A}_2\boldsymbol{b}_V - \mathbf{A}_2\mathbf{W}_V\mathbf{A}_1^{-1}\boldsymbol{v}_1 + \boldsymbol{v}_2}_{=\widetilde{\widetilde{\boldsymbol{b}}}_V} \quad (33)$$

Finally, the linear operation of $V$ has the weights matrix $\widetilde{\widetilde{\mathbf{W}}}_V$ and the bias vector $\widetilde{\widetilde{\boldsymbol{b}}}_V$.

- **Fold $\mathbf{T}_2^{-1}$ into the output projection** Analogous to the folding into $V$, we have incorporated the transformation $\mathbf{T}_1$ into the output projection matrix, as indicated in (31). We now add to it $\mathbf{T}_2^{-1}$, which yields $\widetilde{\widetilde{f}}_O(\boldsymbol{x})$ and is given by:

$$\widetilde{\widetilde{f}}_O(\boldsymbol{x}) = \widetilde{f}_O\left(\mathbf{T}_2^{-1}(\boldsymbol{x})\right) = \underbrace{\mathbf{A}_1\mathbf{W}_O}_{=\widetilde{\mathbf{W}}_O}\mathbf{T}_2^{-1}(\boldsymbol{x}) + \underbrace{\mathbf{A}_1\boldsymbol{b}_O}_{=\widetilde{\boldsymbol{b}}_O} = \underbrace{\mathbf{A}_1\mathbf{W}_O\mathbf{A}_2^{-1}}_{=\widetilde{\widetilde{\mathbf{W}}}_O}\boldsymbol{x} - \underbrace{\mathbf{A}_1\mathbf{W}_O\mathbf{A}_2^{-1}\boldsymbol{v}_2 + \mathbf{A}_1\boldsymbol{b}_O}_{=\widetilde{\widetilde{\boldsymbol{b}}}_O} \quad (34)$$

Finally, the linear operation of the MHA output projections has the weights matrix $\widetilde{\widetilde{\mathbf{W}}}_O$ and the bias vector $\widetilde{\widetilde{\boldsymbol{b}}}_O$.

# D. Implementation Details

The implementation of our algorithm, as well as all evaluated reference methods, is built on top of the microscaling quantization and GPTQ codebase provided by Egiazarian et al. (2025).

The transformations $\mathbf{T}_1$ and $\mathbf{T}_2$ are applied in the same manner as $\mathbf{R}_1$ and $\mathbf{R}_2$ in Liu et al. (2025) and Ashkboos et al. (2024c). The exact mechanism is described in Section 3 of the main paper. All compared reference methods are evaluated using the same transformation structure. In addition, an online Hadamard transformation is applied to the *down* linear operator in the MLP component of each transformer block, following the setup of the cited works. This transformation is not part of the LATMiX algorithm and is included solely to ensure a fair and consistent comparison.

Experiments on models up to 8B parameters were conducted on a single NVIDIA A100 GPU. We used 4 and 8 such GPUs for 14B and 32B parameters models, respectively. For calibration, we used 256 randomly sampled sequences from the WikiText2 dataset (Merity et al., 2016), each with a sequence length of 1,024, for models up to 8B parameters. We used a reduced sequence length of 512 and 128 with an increased number of samples in the same ratio, for the 14B- and 32B-parameter models, respectively. The same calibration set is used both for transformation training and for the GPTQ procedure.

Transformation training is performed using the AdamW optimizer (Loshchilov & Hutter, 2019) with weight-decay, together with a cosine learning-rate scheduler and a linear warmup for 100 steps, with start and end factors of 0.1 and 1, respectively. Optimization is done using 1,000 training steps. The transformation matrix is initialized in a block-diagonal fashion, with small Gaussian noise added to the off-diagonal entries. Each block is a $32 \times 32$ matrix, initialized as a random Hadamard matrix for the LU parametrization or as a random orthogonal matrix for the QR parametrization.

### D.1. Training Hyperparameters

We used the largest batch size permitted by GPU memory for each model, ranging from 1 for the largest models to 8 for smaller ones. The learning rate was tuned separately for each model to ensure stable and effective optimization, with values selected from the range $10^{-3}$ to $10^{-5}$.

In addition, we incorporate two mechanisms into the training procedure to improve stability and perform a sweep to tune their hyperparameters. The first is a regularization term applied to the diagonal entries of the learned transformation matrix, encouraging them to remain close to one. This prevents the diagonal from growing or shrinking excessively, which could otherwise lead to unstable training behavior.

We observe that this regularization has a relatively small but consistent effect. While it does not substantially change the final performance, it helps guide training toward a more stable and better-performing solution. We hypothesize that this limited impact is due to the initialization of the transformation as an orthogonal matrix, whose diagonal entries are already close to one and thus near a good operating point. We used a regularization factor $\lambda = 0.1$ for all model optimizations.

In addition, following common practice in knowledge-distillation-based methods (Hinton et al., 2015), we introduce a calibrated temperature parameter in the distillation loss. Similar to the diagonal regularization, we do not observe a strong sensitivity to the temperature value. However, in several cases it helps the optimization converge to a slightly improved solution.

### D.2. Reference Method Settings

As mentioned earlier, we evaluate a range of existing methods for LLM quantization. Ensuring a fair comparison is challenging, since many of these methods were not originally designed for MX quantization and were evaluated under different quantization formats, transformation choices, and implementation settings. To address this, we re-implement most methods within the same experimental setup and codebase used to evaluate LATMiX, and use the same setup for all methods wherever possible.

Nevertheless, some methods require modifications that depart from their original configurations. For FlatQuant (Sun et al., 2025), a direct one-to-one comparison is infeasible because the original method learns a separate matrix for each block. We therefore aim to remain as faithful as possible to the FlatQuant implementation while keeping the setup close to ours, which also matches common practice in the literature. In particular, we parametrize the learnable matrix $\mathbf{A}_1$ as described in Section 3.1 of the FlatQuant paper, namely as the Kronecker product of two lightweight matrices that are both learned during optimization, but we apply it as a global transformation rather than a per-block one. This baseline also serves to assess the quality of the affine parametrizations proposed in LATMiX, and shows favorable results for our approach. Furthermore, to avoid bias from the choice of objective, we evaluate our implementation of FlatQuant using both its original proposed MSE loss and our distillation-based loss (see Section 8), and report the better result. In practice, the runs using our loss consistently outperform those using the original FlatQuant objective. We further extend the comparison to the fully compressed FlatQuant setting by incorporating per-block (online) transformations together with the corresponding loss terms from the original method. The results for this variant are reported in Table 4.

For SpinQuant, we follow the setup used by (Egiazarian et al., 2025) and therefore do not apply the online Hadamard rotation to the RoPE output. Since these online transformations are orthogonal to the method itself, adding them would likely benefit all methods. As with FlatQuant, we also test SpinQuant using both our distillation-based loss and the original objective used by the method, and report the better result. For SpinQuant, the best results are obtained with cross-entropy loss; the corresponding WikiText2 perplexities for both losses are shown in Table 5.

*Table 4.* Zero-shot accuracy comparison of LATMiX to FlatQuant in its original formulation (marked in bold), which learns an invertible transformation for each transformer block with the associated loss function.

| Method | Llama3.2-1B | Llama3.2-3B-Instruct | Llama3.1-8B-Instruct | Qwen3-1.7B | Qwen3-4B | Qwen3-8B |
|---|---|---|---|---|---|---|
| FlatQuant's matrix structure | 52.45 | 60.55 | 67.81 | 54.85 | 62.46 | 66.89 |
| **FlatQuant** | 52.96 | 60.79 | 67.93 | 55.35 | 62.98 | 67.92 |
| LATMiX | **54.04** | **62.61** | **68.46** | **56.91** | **64.74** | **67.94** |

*Table 5.* Wikitext2 perplexity results for comparison of SpinQuant loss functions.

| Loss Function | Llama3.2-1B | Qwen3-1.7B | Qwen3-8B | Llama3.1-8B-Instruct |
|---|---|---|---|---|
| LATMiX Loss | 13.20 | 26.22 | 11.16 | 8.89 |
| CE | 12.74 | 19.21 | 9.55 | 8.84 |

# E. Additional Experiments

## E.1. Perplexity Results

*Table 6.* Perplexity Comparison on WikiText2 for different quantization methods under MXFP4 quantization for both weights and activations. Best results are highlighted in bold, and second best in underline.

| Method | Llama3.2-1B | Llama3.2-3B | Llama3-8B | Qwen2.5-1.5B | Qwen2.5-3B | Qwen3-1.7B | Qwen3-4B | Qwen3-8B |
|---|---|---|---|---|---|---|---|---|
| FP16 | 9.75 | 7.81 | 6.14 | 9.27 | 8.02 | 16.71 | 13.66 | 9.72 |
| RTN | 17.04 | 10.43 | 8.22 | 15.26 | 12.18 | 22.22 | 17.78 | 11.30 |
| QuaRot-RTN | 21.08 | 13.04 | 11.01 | 14.58 | 11.28 | 38.17 | 20.19 | 13.24 |
| GPTQ | 18.09 | 45.56 | 7.97 | 13.53 | 10.86 | 21.29 | 15.76 | 10.91 |
| QuaRot | 13.05 | 9.56 | 8.08 | 11.63 | 9.62 | 20.00 | 19.26 | 11.14 |
| SpinQuant | 12.67 | 10.08 | 7.76 | 11.39 | 11.35 | 18.10 | 11.97 | 11.21 |
| FlatQuant[†] | 12.79 | 9.66 | 7.81 | 11.53 | 9.60 | 19.28 | 15.10 | 11.02 |
| OSTQuant | 13.30 | 32.30 | 7.86 | – | – | 18.35 | 14.09 | 11.38 |
| BRQ | 11.93 | 9.08 | 7.13 | 11.95 | 9.46 | – | – | – |
| MR-GPTQ | 12.25 | 8.95 | 7.13 | 11.49 | 9.64 | 21.04 | 16.74 | 10.53 |
| **LATMiX-LU (Ours)** | **11.64** | **8.84** | **7.11** | **10.95** | **9.12** | 15.11 | **11.77** | **9.20** |
| **LATMiX-QR (Ours)** | 11.77 | 9.25 | 7.26 | 11.02 | 9.19 | **14.94** | 12.10 | 9.38 |

We evaluate quantization performance on the WikiText2 language modeling benchmark (Merity et al., 2016), using perplexity as the evaluation metric. Table 6 reports MXFP4 results across model families and compares LATMiX with several rotation-based baselines. BRQ denotes the learned block-wise rotation method of Shao et al. (2025b); since BRQ does not report results for Qwen3-1.7B, Qwen3-4B, or Qwen3-8B, we leave the corresponding entries blank. All remaining baseline results are obtained using the publicly released MR-GPTQ codebase (Egiazarian et al., 2025), with adaptations following the transformation setting of SpinQuant (Liu et al., 2025). QuaRot denotes our implementation of tensor-wise Hadamard rotation following Ashkboos et al. (2024c).

Consistent with the trends observed in Section 5, the results show that learning a general affine transformation is more effective than relying on fixed or orthogonal rotation-based transformations. In particular, restricting the learned transformation to be strictly orthogonal, as in BRQ (Shao et al., 2025b), leads to suboptimal performance across models. Allowing a general invertible transformation provides additional flexibility, enabling more effective redistribution of activation energy beyond what is achievable with orthogonal rotations alone.

## E.2. Initialization of the Transformations

*Table 7.* Ablation study on the different methods for initializing the learned transformation matrix, evaluated on WikiText2 perplexity.

| Initialization Method | Llama3.2-1B | |
|---|---|---|
| | LU | QR |
| Identity | 12.23 | 12.12 |
| Identity + Noise | 12.25 | 12.11 |
| Full Orthogonal | 11.86 | 12.35 |
| BD Orthogonal | 12.04 | 11.87 |
| BD Orthogonal + Noise | 12.07 | **11.77** |
| Full Hadamard | 12.27 | 12.14 |
| BD Hadamard | 11.71 | 12.13 |
| BD Hadamard + Noise | **11.64** | 12.09 |

We examined several strategies for initializing the learned transformation matrices. Specifically, we considered initialization as the **Identity** matrix, a random **Orthogonal** matrix, or a random **Hadamard** matrix. For the latter two, we additionally evaluated block-diagonal (**BD**) variants of the initialization. For all diagonal or block-diagonal initializations, we further considered adding small random Gaussian noise to the off-diagonal or off-block-diagonal entries, with the aim of improving optimization stability during training.

The results reported in Table 7 indicate that block-diagonal initialization with small random noise consistently outperforms the other initialization strategies. This behavior aligns with the intuition that a block-diagonal structure is well matched to the MX quantization regime, as it encourages more uniform redistribution of outliers within each quantization block during

the early stages of training. As observed in other ablations in Section 5, the optimization subsequently departs from the initial block-orthogonal structure, enabling controlled transfer of activation energy across blocks and converging to a more effective solution.

### E.3. Loss Function for LATMiX

Here, we justify the choice of the Kullback–Leibler divergence as a measure of the discrepancy between the floating-point and quantized modes. Specifically, let $\boldsymbol{X}$ denote an input context to the model and let Y be a random matrix representing the expected response; we use the expected negative log-likelihood (NLL) as a measure of the model performance (Yang et al., 2019; Li et al., 2025):

$$\mathcal{L}\left(P_\theta\right) \triangleq \mathbb{E}_{\mathrm{Y},\boldsymbol{X}}\left[-\log\left(P_\theta\left(\mathrm{Y}|\boldsymbol{X}\right)\right)\right] = \mathbb{E}_{\boldsymbol{X}}\left[\mathbb{E}_{\mathrm{Y}\sim Q(\cdot|\boldsymbol{X})}\left[-\log\left(P_\theta\left(\mathrm{Y}|\boldsymbol{X}\right)\right)\right]\right], \tag{35}$$

where $P_\theta\left(\mathrm{Y}|\boldsymbol{X}\right)$ represents the probability that the LLM $f$ assigns to each output token (typically computed via a softmax layer), and $Q$ denotes the probability distribution over all possible responses given the context $\boldsymbol{X}$. Let $\widetilde{P}_\Phi$ be the quantized model, parameterized by $\Phi = \{\theta, \Omega\}$, where $\theta$ are the original model parameters and $\Omega$ are the transformation parameters. We analyze the difference between the NNL of the quantized model and that of the full-precision (floating-point) model.

**Proposition E.1.** *Let $\epsilon > 0$ denote a lower bound on the probability values. Suppose that $\epsilon \leq P_\theta\left(\mathrm{Y}|\boldsymbol{X}\right)$ $\quad \forall \mathrm{Y}, \boldsymbol{X}$, and $\epsilon \leq \widetilde{P}_\Phi\left(\mathrm{Y}|\boldsymbol{X}\right)$ $\quad \forall \mathrm{Y}, \boldsymbol{X}$. Then,*

$$\min_\Omega \Delta\left(\Omega\right) \triangleq \min_\Omega\left|\mathcal{L}\left(\widetilde{P}_\Phi\right) - \mathcal{L}\left(P_\theta\right)\right| \leq C + \min_\Omega \mathbb{E}_{\boldsymbol{X}}\left[\mathrm{KL}\left(P_\theta\left(\mathrm{Y}|\boldsymbol{X}\right), \widetilde{P}_\Phi\left(\mathrm{Y}|\boldsymbol{X}\right)\right)\right], \tag{36}$$

*where $C$ is constant independent of $\Omega$.*

The proof is described in Appendix E.3.1. From Proposition E.1, we observe that the mismatch between the quantized model and its floating-point counterpart is captured by an upper bound based on the expected KL divergence over all possible contexts. This establishes a connection between the KL loss and the NLL loss of the downstream task. In practice, however, we typically lack direct access to samples of the downstream task. Yet, it is common to assume access to a corpora of texts and solve a language modeling task instead. Under this assumption, the bound reduces to that in Eq. (8), which is simply the KL divergence between the quantized model and the floating-point model.

Moreover, we investigated three numerical loss functions that are widely used in previous work: (i) MSE applied to the output of each transformer block (Sun et al., 2025) between the quantized and floating-point representations; (ii) cross-entropy loss for the prediction of the next-token (Liu et al., 2025); and (iii) KL divergence between the predictions of the quantized and the floating-point networks. Specifically, we trained an affine transformation with all three losses on Llama3.2-1B and Qwen3-1.7B while quantizing to MXFP4. We report both perplexity on WikiText2 and the average accuracy on the following zero-shot tasks: RC Easy, WinoGrande, PIQA, BoolQ, and OpenBookQA. For all benchmarks, the transformations are learned using WikiText2, making evaluations on WikiText2 in-distribution and zero-shot task evaluations out-of-distribution.

From Table 8, we see that the KL loss outperforms the other loss functions in terms of average accuracy on zero-shot tasks. In addition, while it surpass the MSE loss in terms of WikiText2 perplexity, it yields a higher perplexity compared to the CE loss. This suggests that using CE loss fits better when learning the transformations on the same dataset and task, but worse in zero-shot tasks.

*Table 8.* Ablation study comparing WikiText2 perplexity ($\downarrow$) and zero-shot average accuracy (Avg. Acc. $\uparrow$) for mean squared error (MSE) loss, KL divergence on model predictions, and cross-entropy loss based on next token predictions.

| Loss Function | Llama3.2-1B | | Qwen3-1.7B | |
|---|---|---|---|---|
| | **Wiki** | **Avg. Acc.** | **Wiki** | **Avg. Acc.** |
| FP16 | 9.75 | 60.02 | 16.71 | 63.39 |
| **MSE** | 12.18 | 55.09 | 20.86 | 59.72 |
| **CE** | **11.27** | 56.19 | **13.42** | 58.87 |
| **KL** | 11.64 | **56.89** | 15.11 | **60.86** |

### E.3.1. PROOF PROPOSITION E.1

*Proof.* First, we apply the definition of the NNL loss given in (35)

$$\left| \mathcal{L}\left(\widetilde{P}_\Phi\right) - \mathcal{L}\left(P_\theta\right) \right| = \left| \mathbb{E}_{\boldsymbol{X}}\left[ \mathbb{E}_{Y\sim Q(\cdot|\boldsymbol{X})}\left[ -\log\left( \frac{\widetilde{P}_\Phi(Y|\boldsymbol{X})}{P_\theta(Y|\boldsymbol{X})} \right) \right] \right] \right|. \tag{37}$$

Next, we insert and subtract the floating-point model probability $P_\theta\left(Y|\boldsymbol{X}\right)$ inside the expectation, which gives:

$$I_1 = \mathbb{E}_{Y\sim Q(\cdot|\boldsymbol{X})}\left[ -\log\left( \frac{\widetilde{P}_\Phi(Y|\boldsymbol{X})}{P_\theta(Y|\boldsymbol{X})} \right) \right] = \underbrace{\mathbb{E}_{Y\sim P_\theta(\cdot|\boldsymbol{X})}\left[ -\log\left( \frac{\widetilde{P}_\Phi(Y|\boldsymbol{X})}{P_\theta(Y|\boldsymbol{X})} \right) \right]}_{\mathrm{KL}\left(P_\theta(Y|\boldsymbol{X}),\widetilde{P}_\Phi(Y|\boldsymbol{X})\right)} + I_2 \quad , \text{ where} \tag{38}$$

$$I_2 = \mathbb{E}_{Y\sim Q(\cdot|\boldsymbol{X})}\left[ -\log\left( \frac{\widetilde{P}_\Phi(Y|\boldsymbol{X})}{P_\theta(Y|\boldsymbol{X})} \right) \right] - \mathbb{E}_{Y\sim P_\theta(\cdot|\boldsymbol{X})}\left[ -\log\left( \frac{\widetilde{P}_\Phi(Y|\boldsymbol{X})}{P_\theta(Y|\boldsymbol{X})} \right) \right].$$

By defining $g\left(Y\right) = -\log\left( \frac{\widetilde{P}_\Phi(Y|\boldsymbol{X})}{P_\theta(Y|\boldsymbol{X})} \right)$ we have

$$|I_2| \leq \sum_Y |g\left(Y\right)| \left| Q\left(Y|\boldsymbol{X}\right) - P_\theta\left(Y|\boldsymbol{X}\right) \right| \leq \|g\|_\infty \sum_Y \left| Q\left(Y|\boldsymbol{X}\right) - P_\theta\left(Y|\boldsymbol{X}\right) \right|$$

$$\leq 2\|g\|_\infty \,\mathrm{TV}\left( Q\left(\cdot|\boldsymbol{X}\right), P_\theta\left(\cdot|\boldsymbol{X}\right) \right), \tag{39}$$

where $\|g\|_\infty \triangleq \sup_{\boldsymbol{y}}|g\left(\boldsymbol{y}\right)|$ is the infinite norm of the function $g$ and $\mathrm{TV}\left( Q\left(\cdot|\boldsymbol{X}\right), P_\theta\left(\cdot|\boldsymbol{X}\right) \right) \triangleq \frac{1}{2}\sum_Y \left| Q\left(Y|\boldsymbol{X}\right) - P_\theta\left(Y|\boldsymbol{X}\right) \right|$ is the total variation between the distribution $Q$ and $P$. Using $\epsilon < Q\left(\cdot|\boldsymbol{X}\right)$ and $\epsilon < P_\theta\left(\cdot|\boldsymbol{X}\right)$ we have $\|g\|_\infty \leq \log\left( \frac{1-\epsilon}{\epsilon} \right)$ resulting in:

$$|I_1| \leq \mathrm{KL}\left( P_\theta\left(Y|\boldsymbol{X}\right), \widetilde{P}_\Phi\left(Y|\boldsymbol{X}\right) \right) + 2\log\left( \frac{1-\epsilon}{\epsilon} \right) \mathrm{TV}\left( Q\left(Y|\boldsymbol{X}\right), P_\theta\left(Y|\boldsymbol{X}\right) \right) \tag{40}$$

Finally, combining (37),(38) and (40) yields the desired result.

$\square$

### E.4. Transformation Parametrization Condition Number

Figure 6 tracks the condition number of the learned $\mathbf{A}_1$ transformation during training for both LU and QR parameterizations. This experiment verifies that the additional flexibility of affine transformations does not lead to numerically unstable matrices. Both parameterizations remain well-conditioned throughout optimization, supporting the use of learned invertible transformations in the folding procedure.

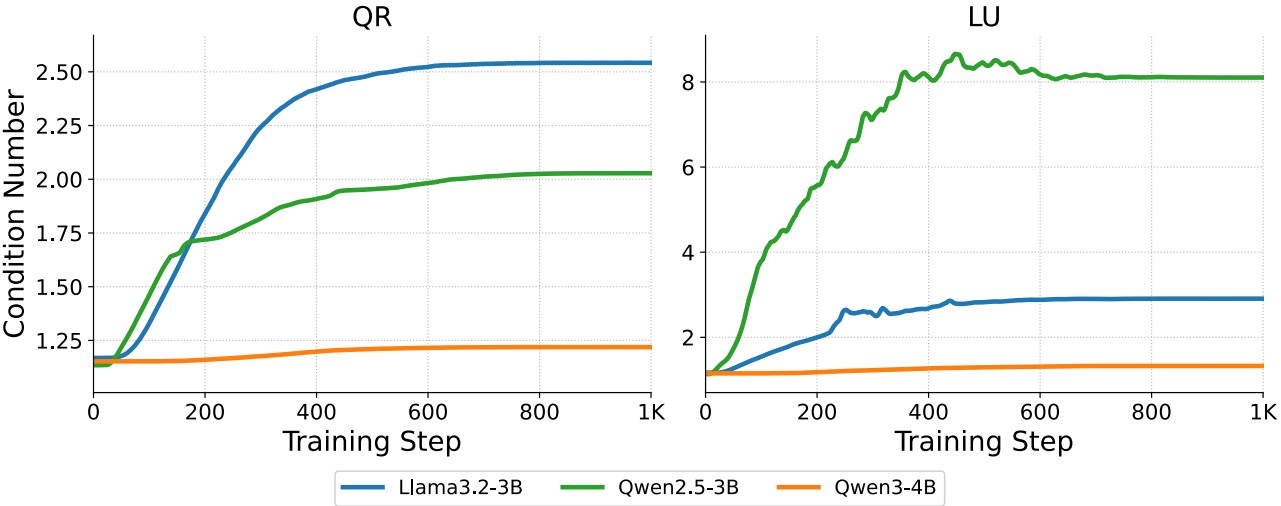

*Figure 6.* $\mathbf{A}_1$ matrix condition number throughout the training for the QR and LU matrix parametrization.

### E.5. Extended Ablations

#### E.5.1. CALIBRATION SET SIZE

Table 9 evaluates the dependence of LATMiX on the number of calibration samples used to learn the transformations. Since the method is intended as a post-training procedure, it is important that the learned affine transformations can be estimated from a small, unlabeled calibration set. The results show that performance improves rapidly with only a few samples and remains stable as the set grows, indicating that the transformation learning is data-efficient and does not require a large calibration corpus.

*Table 9.* Wikitext2 perplexity ($\downarrow$) and Zero-shot ($\uparrow$) accuracy results with varying number of calibration samples selected randomly from the WikiText2 training set.

| Set Size | Llama-3.1-8B-Instruct | | Qwen3-8B | |
|---|---|---|---|---|
| | Wiki | Avg. Acc. | Wiki | Avg. Acc. |
| 1 | 8.93 | 69.11 | 11.03 | 66.33 |
| 4 | 8.53 | 70.11 | 10.00 | 67.24 |
| 8 | 8.38 | 70.41 | 9.71 | 67.82 |
| 64 | 8.04 | 70.32 | 9.52 | 68.16 |
| 128 | 8.03 | 70.31 | 9.35 | 68.12 |
| 256 | 7.97 | 70.51 | 9.38 | 68.46 |
| 512 | 8.07 | 71.99 | 9.40 | 69.68 |

### E.5.2. RANDOM CALIBRATION SET SELECTION

Table 10 evaluates sensitivity to the particular calibration subset by repeating transformation learning over several random WikiText2 subsets. The small standard deviations across tasks indicate that LATMiX is not strongly dependent on a favorable calibration draw. This supports the view that the learned transformations capture stable activation-structure properties of the model rather than overfitting to a specific set of examples.

*Table 10.* Robustness to the choice of calibration subset. Presenting the mean and STD of five zero-shot benchmarks over the selection of five different random subsets of Wikitext2 for calibration.

| Benchmark / Model | Llama-3.1-8B-Instruct | Qwen3-8B |
|---|---|---|
| ARC-E | $72.85 \pm 0.97$ | $76.55 \pm 1.15$ |
| BoolQ | $83.37 \pm 0.64$ | $84.57 \pm 0.84$ |
| WinoGrande | $72.71 \pm 0.89$ | $68.72 \pm 0.70$ |
| PIQA | $78.26 \pm 0.63$ | $76.14 \pm 0.58$ |
| OBQA | $45.22 \pm 1.48$ | $40.54 \pm 0.91$ |
| Avg. Accuracy | $70.48 \pm 0.35$ | $69.31 \pm 0.42$ |
| Avg. Recovery (%) | $96.37 \pm 0.62$ | $97.18 \pm 0.63$ |

### E.5.3. NUMBER OF TRAINING STEPS

Table 11 studies the effect of the number of optimization steps used to learn the transformations. The comparison shows that most of the gain is obtained early in training, with strong performance already achieved after a relatively small number of steps. This indicates that LATMiX can learn useful affine transformations quickly, keeping the additional training overhead of the method modest.

*Table 11.* Number of optimization steps ablation on Llama-3.1-8B-Instruct.

| Training steps | PPL $\downarrow$ | Avg. 0-Shot $\uparrow$ |
|---|---|---|
| 0 | 8.13 | 64.35 |
| 100 | 8.07 | 67.48 |
| 200 | 8.05 | 68.02 |
| 500 | 8.01 | 68.55 |
| 750 | 8.02 | 68.96 |
| **1000** | 7.97 | 68.46 |
| 2500 | 8.03 | 68.12 |
| 5000 | 8.02 | 68.94 |

### E.5.4. REGULARIZATION FACTOR

Table 12 reports the sensitivity to the volume-preserving regularization coefficient. This regularizer is used to keep the learned transformation well-behaved while still allowing individual directions to expand or contract. The results show that performance is relatively stable across a broad range of coefficients, with $\lambda = 0.1$ providing a strong operating point across both perplexity and zero-shot accuracy.

*Table 12.* Sensitivity to the regularization coefficient $\lambda$ on Qwen3-1.7B.

| $\lambda$ | Wiki $\downarrow$ | Avg. Acc. $\uparrow$ |
|---|---|---|
| 0.001 | 17.71 | 56.73 |
| 0.01 | 17.67 | 56.23 |
| 0.05 | 17.32 | 56.36 |
| **0.1** | 17.37 | 56.91 |
| 0.5 | 17.40 | 56.15 |
| 1 | 18.18 | 56.41 |
| 10 | 17.90 | 56.57 |

### E.5.5. LOSS TEMPERATURE

Table 13 examines the effect of the softmax temperature used in the distillation objective. The temperature controls how much information from the teacher's distribution is exposed during transformation learning. Overall, the results are relatively insensitive to the exact temperature over a moderate range, although tuning this parameter can help select a better operating point for a given model.

*Table 13.* Sensitivity to the softmax temperature on Llama-3.1-8B-Instruct.

| Temperature | Wiki $\downarrow$ | Avg. Acc. $\uparrow$ |
|---|---|---|
| 0.1 | 38.20 | 68.30 |
| 0.5 | 8.47 | 68.89 |
| 0.75 | 8.09 | 68.19 |
| 1 | 8.04 | 68.37 |
| **1.5** | 7.97 | 68.46 |
| 2 | 8.09 | 68.47 |
| 5 | 8.08 | 67.87 |

### E.5.6. SINGLE TRANSFORM ABLATION

Table 14 shows the contribution of each transformation to WikiText2 perplexity. The results demonstrate the benefit of applying all three transformations, as removing any one of them consistently degrades perplexity across the evaluated models.

*Table 14.* Wikitext2 perplexity without each of the three transformations applied by LATMiX.

| Model | All | No $T_3$ | No $T_1$ | No $T_2$ |
|---|---|---|---|---|
| Llama3.1-8B-Instruct | 7.97 | 8.14 | 8.17 | 8.07 |
| Qwen3-1.7B | 15.72 | 15.87 | 17.38 | 18.16 |
| Qwen3-8B | 9.38 | 9.54 | 9.60 | 9.47 |

### E.6. NVFP Quantization Format Evaluation

In Table 15, we compare LATMiX against leading approaches under NVFP quantization. Across most comparisons, LATMiX attains the best or second-best performance, indicating that the learned transformations generalize well to this quantization setup as well.

*Table 15.* NVFP quantization results

| Model | Method | ARC-C | ARC-E | HellaSwag | WinoGrande | PIQA | BOOLQ | OBQA | Avg. Accuracy | Avg. Recovery (%) |
|---|---|---|---|---|---|---|---|---|---|---|
| **Llama3.2-1B** | RTN | 31.91 | 57.53 | 60.92 | 58.72 | 70.78 | 62.23 | 34.00 | 53.73 | 92.86 |
| | GPTQ | 33.62 | 57.24 | 61.62 | 59.67 | 70.13 | 59.60 | 33.80 | 53.67 | 93.03 |
| | SpinQuant | 34.64 | 57.45 | 60.52 | 59.35 | 71.49 | 61.62 | 35.40 | 54.35 | 94.49 |
| | FlatQuant† | 33.96 | 55.85 | 59.85 | 58.72 | 70.18 | 61.93 | 34.20 | 53.53 | 92.93 |
| | MR-GPTQ | 32.85 | 58.04 | 60.06 | 55.96 | 71.55 | 59.51 | 35.20 | 53.31 | 92.52 |
| | **LATMiX-LU (Ours)** | 35.24 | 59.47 | 60.95 | 59.43 | 72.25 | 61.41 | 35.80 | **54.94** | **95.56** |
| | **LATMiX-QR (Ours)** | 33.28 | 59.34 | 61.77 | 59.43 | 72.09 | 61.77 | 35.40 | 54.73 | 94.84 |
| **Llama3.2-3B-Instruct** | RTN | 42.75 | 65.87 | 69.08 | 66.14 | 74.65 | 74.86 | 41.60 | 62.13 | 97.79 |
| | GPTQ | 42.32 | 65.66 | 69.93 | 65.59 | 74.86 | 76.12 | 40.40 | 62.12 | 97.53 |
| | SpinQuant | 43.17 | 62.96 | 70.57 | 64.48 | 74.10 | 73.30 | 40.20 | 61.26 | 96.37 |
| | FlatQuant† | 40.96 | 60.10 | 69.80 | 66.38 | 73.34 | 74.01 | 38.20 | 60.40 | 94.62 |
| | MR-GPTQ | 41.89 | 67.30 | 70.13 | 64.80 | 74.54 | 78.23 | 40.40 | **62.47** | **97.95** |
| | **LATMiX-LU (Ours)** | 43.77 | 65.82 | 68.83 | 67.40 | 73.88 | 78.50 | 38.80 | 62.43 | 97.78 |
| | **LATMiX-QR (Ours)** | 41.89 | 63.09 | 70.08 | 66.46 | 74.32 | 74.65 | 40.00 | 61.50 | 96.42 |
| **Llama3.1-8B-Instruct** | RTN | 51.37 | 74.37 | 77.72 | 72.93 | 78.45 | 81.68 | 47.80 | 69.19 | 97.50 |
| | GPTQ | 51.88 | 74.66 | 78.50 | 71.74 | 78.78 | 83.15 | 47.80 | **69.50** | **97.92** |
| | SpinQuant | 51.62 | 73.32 | 77.54 | 71.51 | 78.56 | 82.23 | 46.20 | 68.71 | 96.72 |
| | FlatQuant† | 51.96 | 74.79 | 77.48 | 72.22 | 78.56 | 81.44 | 45.40 | 68.84 | 96.84 |
| | MR-GPTQ | 51.11 | 75.04 | 77.69 | 75.06 | 79.87 | 81.62 | 46.40 | 69.54 | 97.79 |
| | **LATMiX-LU (Ours)** | 52.05 | 73.70 | 77.38 | 75.06 | 79.16 | 83.49 | 45.20 | 69.43 | 97.57 |
| | **LATMiX-QR (Ours)** | 50.94 | 73.70 | 77.53 | 74.51 | 79.16 | 84.62 | 46.20 | 69.52 | 97.68 |
| **Qwen3-1.7B** | RTN | 37.63 | 61.20 | 58.05 | 60.54 | 70.18 | 74.71 | 34.80 | 56.73 | 94.12 |
| | GPTQ | 37.46 | 63.17 | 56.68 | 58.56 | 69.21 | 76.94 | 33.80 | 56.55 | 93.52 |
| | SpinQuant | 34.56 | 54.88 | 56.07 | 56.27 | 68.72 | 73.70 | 34.00 | 54.03 | 89.55 |
| | FlatQuant† | 36.35 | 56.99 | 55.62 | 57.85 | 68.06 | 73.94 | 33.00 | 54.54 | 90.37 |
| | MR-GPTQ | 38.99 | 63.22 | 56.98 | 57.38 | 69.64 | 72.23 | 37.40 | 56.55 | 94.43 |
| | **LATMiX-LU (Ours)** | 39.08 | 60.44 | 58.94 | 59.19 | 69.91 | 76.64 | 36.40 | 57.23 | 95.26 |
| | **LATMiX-QR (Ours)** | 40.10 | 64.18 | 58.48 | 60.06 | 70.13 | 77.43 | 34.80 | **57.89** | **96.04** |
| **Qwen3-4B** | RTN | 48.38 | 73.19 | 67.81 | 65.11 | 74.10 | 82.75 | 37.60 | 64.13 | 95.36 |
| | GPTQ | 49.74 | 73.65 | 67.98 | 65.51 | 73.12 | 82.97 | 38.20 | 64.45 | 95.99 |
| | SpinQuant | 44.37 | 68.90 | 64.67 | 62.75 | 71.33 | 83.24 | 36.00 | 61.61 | 91.34 |
| | FlatQuant† | 46.67 | 69.44 | 65.96 | 62.43 | 72.25 | 83.43 | 38.80 | 62.71 | 93.46 |
| | MR-GPTQ | 47.95 | 70.66 | 65.88 | 64.09 | 71.60 | 83.03 | 37.00 | 62.89 | 93.53 |
| | **LATMiX-LU (Ours)** | 47.95 | 74.12 | 69.15 | 65.67 | 75.14 | 83.49 | 39.00 | 64.93 | 96.62 |
| | **LATMiX-QR (Ours)** | 48.55 | 73.91 | 68.90 | 65.59 | 74.32 | 83.64 | 39.60 | **64.93** | **96.76** |
| **Qwen3-8B** | RTN | 55.20 | 78.83 | 74.44 | 68.51 | 76.06 | 86.09 | 41.60 | 68.68 | 98.17 |
| | GPTQ | 55.20 | 79.55 | 74.29 | 67.01 | 76.39 | 86.36 | 41.80 | 68.66 | 98.13 |
| | SpinQuant | 53.75 | 77.40 | 73.05 | 68.03 | 75.08 | 85.75 | 39.60 | 67.52 | 96.27 |
| | FlatQuant† | 51.79 | 78.75 | 73.23 | 67.56 | 76.50 | 85.84 | 42.00 | 67.95 | 97.05 |
| | MR-GPTQ | 55.12 | 78.91 | 74.22 | 68.59 | 76.33 | 85.41 | 41.60 | 68.60 | 98.07 |
| | **LATMiX-LU (Ours)** | 55.38 | 78.41 | 75.60 | 70.64 | 77.64 | 86.24 | 42.00 | **69.41** | **99.24** |
| | **LATMiX-QR (Ours)** | 53.24 | 77.82 | 73.59 | 67.56 | 76.28 | 85.23 | 42.00 | 67.96 | 97.18 |

# F. Full Experimental Results

Here we provide the complete benchmark breakdown corresponding to the results summarized in Table 1 for Llama-3.2-1B, Llama-3.2-3B-Instruct, Llama-3.1-8B-Instruct, Qwen3-1.7B, Qwen3-4B, Qwen3-8B, Qwen3-14B, and Qwen3-32B. For each model, we report per-task zero-shot accuracies on ARC-Challenge, ARC-Easy, HellaSwag, WinoGrande, PIQA, BoolQ, and OpenBookQA under both MXFP4 and MXINT4 weight and activation quantization. These tables provide the task-level results behind the average accuracy and average recovery values reported in the main text.

We compare LATMiX to RTN, QuaRot-RTN, GPTQ, QuaRot, SpinQuant, OSTQuant, FlatQuant[†], and MR-GPTQ under the same experimental setup. All methods except RTN and QuaRot-RTN are combined with GPTQ. As in the main experiments, FlatQuant[†] denotes using FlatQuant's matrix structure in our experimental setup, as detailed in Appendix D.2. For both LATMiX and FlatQuant[†], the transformations were learned using the WikiText2 dataset.

*Table 16.* Performance of Llama-3.2-1B across weight and activation quantization configurations.

| Format | Method | ARC-C | ARC-E | HellaSwag | WinoGrande | PIQA | BOOLQ | OBQA | Avg. Accuracy | Avg. Recovery (%) |
|---|---|---|---|---|---|---|---|---|---|---|
| | FP16 | 36.86 | 61.70 | 65.79 | 62.75 | 74.81 | 63.82 | 37.00 | 57.53 | 100 |
| MXFP4 | RTN | 30.38 | 49.79 | 51.66 | 55.01 | 67.68 | 55.11 | 31.80 | 48.78 | 84.58 |
| | QuaRot-RTN | 27.13 | 45.03 | 46.88 | 52.57 | 64.96 | 58.75 | 29.40 | 46.39 | 80.00 |
| | GPTQ | 30.20 | 51.30 | 54.37 | 53.91 | 67.19 | 55.57 | 32.00 | 49.22 | 85.29 |
| | QuaRot | 31.74 | 55.68 | 57.03 | 58.48 | 69.10 | 56.91 | 33.20 | 51.74 | 89.64 |
| | SpinQuant | 30.72 | 53.41 | 56.54 | 54.78 | 69.59 | 61.22 | 36.40 | 51.81 | 90.06 |
| | OSTQuant | 31.82 | 56.94 | 56.61 | 56.35 | 69.69 | 61.86 | 34.80 | 52.58 | 91.22 |
| | FlatQuant[†] | 30.89 | 55.56 | 58.57 | 57.93 | 69.26 | 62.14 | 32.80 | 52.45 | 90.54 |
| | MR-GPTQ | 34.30 | 57.32 | 57.53 | 59.51 | 69.70 | 61.65 | 34.60 | 53.52 | 93.07 |
| | **LATMiX-LU (Ours)** | 34.72 | 58.50 | 59.09 | 58.40 | 70.89 | 62.29 | 34.40 | **54.04** | **93.88** |
| | **LATMiX-QR (Ours)** | 34.04 | 58.54 | 57.50 | 58.40 | 71.16 | 62.32 | 34.60 | 53.79 | 93.42 |
| MXINT4 | RTN | 27.22 | 43.22 | 43.35 | 54.30 | 61.92 | 48.65 | 29.20 | 43.98 | 76.32 |
| | QuaRot-RTN | 29.35 | 44.19 | 43.38 | 52.41 | 59.30 | 54.80 | 29.40 | 44.69 | 77.90 |
| | GPTQ | 25.94 | 43.73 | 46.72 | 50.75 | 60.45 | 50.89 | 26.80 | 43.61 | 75.16 |
| | QuaRot | 31.83 | 52.02 | 54.39 | 53.99 | 66.10 | 57.19 | 33.80 | 49.90 | 86.95 |
| | SpinQuant | 29.10 | 48.78 | 53.53 | 55.17 | 65.67 | 61.56 | 33.00 | 49.54 | 85.81 |
| | OSTQuant | 30.72 | 54.04 | 53.90 | 54.38 | 67.46 | 59.11 | 33.40 | 50.43 | 87.51 |
| | FlatQuant[†] | 31.14 | 54.12 | 56.04 | 57.46 | 67.41 | 54.01 | 32.20 | 50.34 | 87.25 |
| | MR-GPTQ | 31.23 | 52.10 | 57.22 | 57.46 | 68.44 | 59.63 | 31.40 | 51.07 | 88.21 |
| | **LATMiX-LU (Ours)** | 31.99 | 54.46 | 57.25 | 58.24 | 69.69 | 61.43 | 32.60 | 52.24 | 90.34 |
| | **LATMiX-QR (Ours)** | 32.08 | 53.82 | 56.92 | 56.19 | 69.58 | 61.95 | 36.20 | **52.39** | **91.17** |

*Table 17.* Performance of Llama3.2-3B-Instruct across weight and activation quantization configurations.

| Format | Method | ARC-C | ARC-E | HellaSwag | WinoGrande | PIQA | BOOLQ | OBQA | Avg. Accuracy | Avg. Recovery (%) |
|---|---|---|---|---|---|---|---|---|---|---|
| | FP16 | 45.56 | 67.42 | 71.72 | 67.32 | 75.41 | 75.35 | 42.00 | 63.54 | 100 |
| MXFP4 | RTN | 43.00 | 60.65 | 67.43 | 62.59 | 73.18 | 73.76 | 39.20 | 59.97 | 94.06 |
| | QuaRot-RTN | 33.53 | 48.44 | 62.04 | 58.64 | 71.60 | 71.50 | 34.00 | 54.25 | 84.13 |
| | GPTQ | 41.21 | 61.87 | 67.25 | 62.83 | 71.98 | 73.03 | 37.80 | 59.42 | 92.92 |
| | QuaRot | 41.04 | 56.31 | 68.19 | 65.04 | 71.16 | 72.66 | 36.60 | 58.72 | 91.74 |
| | SpinQuant | 39.93 | 63.30 | 66.12 | 62.12 | 72.52 | 75.60 | 38.20 | 59.68 | 93.17 |
| | OSTQuant | 40.18 | 61.82 | 66.52 | 63.45 | 72.14 | 74.46 | 36.60 | 59.31 | 92.64 |
| | FlatQuant[†] | 40.44 | 63.09 | 68.45 | 64.48 | 74.32 | 74.28 | 38.80 | 60.55 | 94.73 |
| | MR-GPTQ | 43.17 | 65.45 | 69.06 | 64.17 | 73.50 | 70.58 | 38.60 | 60.65 | 95.02 |
| | **LATMiX-LU (Ours)** | 43.43 | 67.46 | 69.59 | 64.24 | 74.59 | 79.93 | 39.00 | **62.61** | **97.95** |
| | **LATMiX-QR (Ours)** | 42.57 | 65.26 | 67.72 | 63.77 | 73.88 | 78.04 | 40.00 | 61.61 | 96.59 |
| MXINT4 | RTN | 36.01 | 55.77 | 64.04 | 59.43 | 70.57 | 67.74 | 37.60 | 55.88 | 87.31 |
| | QuaRot-RTN | 33.28 | 48.15 | 56.00 | 58.17 | 66.05 | 57.92 | 31.00 | 50.08 | 78.02 |
| | GPTQ | 35.24 | 54.42 | 65.06 | 62.35 | 70.73 | 70.09 | 33.80 | 55.96 | 86.79 |
| | QuaRot | 39.42 | 56.86 | 67.02 | 60.38 | 71.82 | 69.57 | 38.00 | 57.58 | 90.13 |
| | SpinQuant | 41.47 | 62.96 | 67.27 | 65.35 | 72.52 | 72.78 | 37.80 | 60.02 | 93.83 |
| | OSTQuant | 37.20 | 55.98 | 64.65 | 63.46 | 72.42 | 71.83 | 38.40 | 57.70 | 90.11 |
| | FlatQuant[†] | 39.25 | 58.54 | 66.06 | 63.93 | 72.03 | 74.13 | 34.20 | 58.31 | 90.77 |
| | MR-GPTQ | 41.98 | 64.02 | 69.03 | 63.14 | 73.83 | 72.54 | 40.20 | 60.68 | 95.11 |
| | **LATMiX-LU (Ours)** | 41.97 | 64.85 | 67.99 | 64.71 | 73.83 | 73.82 | 38.20 | 60.77 | 95.15 |
| | **LATMiX-QR (Ours)** | 41.63 | 64.85 | 66.89 | 66.29 | 73.44 | 75.10 | 40.00 | **61.17** | **95.94** |

*Table 18.* Performance of Llama-3.1-8B-Instruct across weight and activation quantization configurations.

| Format | Method | ARC-C | ARC-E | HellaSwag | WinoGrande | PIQA | BOOLQ | OBQA | Avg. Accuracy | Avg. Recovery (%) |
|---|---|---|---|---|---|---|---|---|---|---|
| | FP16 | 53.58 | 75.76 | 78.51 | 76.09 | 79.49 | 83.88 | 49.00 | 70.90 | 100 |
| MXFP4 | RTN | 47.87 | 72.18 | 74.85 | 70.24 | 76.88 | 82.17 | 43.20 | 66.77 | 93.59 |
| | QuaRot-RTN | 43.34 | 67.34 | 71.05 | 61.80 | 74.05 | 78.38 | 41.00 | 62.42 | 87.40 |
| | GPTQ | 48.98 | 73.48 | 76.07 | 69.77 | 78.67 | 80.28 | 44.80 | 67.44 | 94.73 |
| | QuaRot | 44.62 | 70.50 | 75.20 | 70.96 | 76.99 | 83.55 | 44.80 | 66.66 | 93.32 |
| | SpinQuant | 48.98 | 74.66 | 75.74 | 72.30 | 77.48 | 82.75 | 44.80 | 68.10 | 95.57 |
| | OSTQuant | 48.37 | 73.77 | 74.86 | 69.29 | 77.47 | 81.71 | 43.80 | 67.03 | 94.04 |
| | FlatQuant[†] | 50.09 | 72.73 | 75.22 | 71.03 | 77.75 | 82.08 | 45.80 | 67.81 | 95.40 |
| | MR-GPTQ | 48.98 | 73.48 | 76.07 | 69.77 | 78.67 | 80.28 | 44.80 | 67.44 | 94.73 |
| | **LATMiX-LU (Ours)** | 49.82 | 73.02 | 76.53 | 72.53 | 78.40 | 83.30 | 45.60 | **68.46** | **96.17** |
| | **LATMiX-QR (Ours)** | 48.72 | 74.41 | 76.55 | 71.90 | 78.07 | 82.41 | 47.00 | 68.44 | 96.21 |
| MXINT4 | RTN | 45.90 | 70.16 | 72.31 | 67.48 | 76.77 | 77.09 | 42.60 | 64.62 | 90.64 |
| | QuaRot-RTN | 41.64 | 61.11 | 71.04 | 59.19 | 72.74 | 70.18 | 39.40 | 59.33 | 83.18 |
| | GPTQ | 44.45 | 67.63 | 71.16 | 65.43 | 76.88 | 79.57 | 40.60 | 63.68 | 89.04 |
| | QuaRot | 44.62 | 70.50 | 75.20 | 70.96 | 76.99 | 83.55 | 44.80 | 66.66 | 93.32 |
| | SpinQuant | 48.98 | 74.66 | 75.74 | 72.30 | 77.48 | 82.75 | 44.80 | 68.10 | 95.57 |
| | OSTQuant | 46.50 | 69.36 | 74.93 | 71.51 | 75.95 | 79.63 | 45.00 | 66.13 | 92.87 |
| | FlatQuant[†] | 47.61 | 69.15 | 75.02 | 68.67 | 78.02 | 83.18 | 39.00 | 65.81 | 91.84 |
| | MR-GPTQ | 52.56 | 76.22 | 77.44 | 74.43 | 78.67 | 80.76 | 42.40 | 68.93 | 96.71 |
| | **LATMiX-LU (Ours)** | 49.23 | 73.06 | 76.63 | 72.92 | 78.18 | 82.69 | 46.40 | 68.44 | **96.20** |
| | **LATMiX-QR (Ours)** | 49.14 | 75.21 | 75.80 | 71.03 | 78.29 | 83.57 | 44.00 | 68.15 | 95.54 |

*Table 19.* Performance of Qwen3-1.7B across weight and activation quantization configurations.

| Format | Method | ARC-C | ARC-E | HellaSwag | WinoGrande | PIQA | BOOLQ | OBQA | Avg. Accuracy | Avg. Recovery (%) |
|---|---|---|---|---|---|---|---|---|---|---|
| | FP16 | 43.17 | 69.57 | 60.17 | 60.38 | 72.20 | 77.61 | 37.20 | 60.04 | 100 |
| MXFP4 | RTN | 34.13 | 54.42 | 52.02 | 56.43 | 65.13 | 66.21 | 31.40 | 51.39 | 85.30 |
| | QuaRot-RTN | 29.35 | 45.45 | 46.33 | 51.62 | 61.10 | 65.60 | 27.60 | 46.72 | 77.02 |
| | GPTQ | 30.55 | 50.13 | 51.60 | 57.54 | 66.87 | 66.48 | 33.80 | 50.99 | 84.71 |
| | QuaRot | 34.73 | 51.18 | 54.45 | 53.75 | 67.52 | 66.64 | 34.40 | 51.81 | 86.48 |
| | SpinQuant | 34.90 | 53.45 | 54.30 | 59.12 | 66.16 | 70.40 | 30.60 | 52.70 | 87.20 |
| | OSTQuant | 38.13 | 63.55 | 54.43 | 58.56 | 67.57 | 71.16 | 35.00 | 55.48 | 92.35 |
| | FlatQuant† | 36.69 | 57.58 | 56.04 | 57.06 | 68.72 | 74.07 | 33.80 | 54.85 | 90.98 |
| | MR-GPTQ | 33.36 | 54.67 | 53.08 | 57.22 | 65.89 | 69.02 | 32.40 | 52.23 | 86.59 |
| | **LATMiX-LU (Ours)** | 40.27 | 62.62 | 57.26 | 60.29 | 69.04 | 76.17 | 36.20 | **57.42** | **95.62** |
| | **LATMiX-QR (Ours)** | 38.31 | 63.51 | 56.48 | 59.43 | 69.64 | 76.20 | 33.20 | 56.68 | 93.74 |
| MXINT4 | RTN | 28.67 | 44.74 | 45.51 | 54.70 | 61.92 | 60.09 | 29.20 | 46.40 | 76.94 |
| | QuaRot-RTN | 25.94 | 35.02 | 40.00 | 50.67 | 56.26 | 61.47 | 26.20 | 42.22 | 69.77 |
| | GPTQ | 28.84 | 44.40 | 46.75 | 52.33 | 62.30 | 58.87 | 29.80 | 46.18 | 76.75 |
| | QuaRot | 32.68 | 44.87 | 54.27 | 54.78 | 64.15 | 65.72 | 31.60 | 49.72 | 82.80 |
| | SpinQuant | 29.52 | 48.48 | 50.59 | 56.04 | 64.42 | 68.44 | 29.20 | 49.53 | 81.55 |
| | OSTQuant | 38.31 | 57.03 | 55.41 | 56.59 | 67.63 | 72.02 | 33.60 | 54.37 | 90.47 |
| | FlatQuant† | 34.47 | 51.68 | 54.57 | 58.64 | 67.36 | 69.91 | 35.20 | 53.12 | 88.56 |
| | MR-GPTQ | 35.41 | 51.56 | 54.52 | 55.64 | 66.05 | 66.64 | 32.60 | 51.77 | 86.27 |
| | **LATMiX-LU (Ours)** | 37.37 | 59.80 | 57.32 | 55.88 | 69.15 | 72.90 | 35.60 | 55.43 | 92.24 |
| | **LATMiX-QR (Ours)** | 37.62 | 62.28 | 55.25 | 59.35 | 68.60 | 72.38 | 35.60 | **55.87** | **92.96** |

*Table 20.* Performance of Qwen3-4B across weight and activation quantization configurations.

| Format | Method | ARC-C | ARC-E | HellaSwag | WinoGrande | PIQA | BOOLQ | OBQA | Avg. Accuracy | Avg. Recovery (%) |
|---|---|---|---|---|---|---|---|---|---|---|
| | FP16 | 53.50 | 78.41 | 70.02 | 67.01 | 74.97 | 85.02 | 40.20 | 67.02 | 100 |
| MXFP4 | RTN | 45.14 | 67.00 | 64.24 | 61.01 | 71.11 | 74.80 | 35.00 | 59.76 | 88.93 |
| | QuaRot-RTN | 39.33 | 57.24 | 57.41 | 61.64 | 69.15 | 70.64 | 36.60 | 56.00 | 83.84 |
| | GPTQ | 46.50 | 68.52 | 65.44 | 63.85 | 72.03 | 80.86 | 37.60 | 62.11 | 92.54 |
| | QuaRot | 46.16 | 70.79 | 62.88 | 62.90 | 70.24 | 82.32 | 36.00 | 61.61 | 91.47 |
| | SpinQuant | 43.94 | 65.91 | 63.65 | 60.22 | 71.82 | 79.02 | 38.00 | 60.37 | 90.04 |
| | OSTQuant | 45.90 | 70.03 | 65.29 | 63.61 | 72.47 | 80.55 | 39.00 | 62.40 | 93.10 |
| | FlatQuant† | 46.84 | 70.37 | 64.87 | 61.96 | 72.63 | 82.78 | 37.80 | 62.46 | 92.96 |
| | MR-GPTQ | 46.50 | 68.52 | 65.44 | 63.85 | 72.03 | 80.86 | 37.60 | 62.11 | 92.54 |
| | **LATMiX-LU (Ours)** | 48.29 | 73.95 | 64.84 | 63.61 | 72.74 | 83.33 | 39.40 | **64.74** | **96.64** |
| | **LATMiX-QR (Ours)** | 48.55 | 74.07 | 65.98 | 63.38 | 73.67 | 83.73 | 38.80 | 64.08 | 95.64 |
| MXINT4 | RTN | 36.09 | 50.72 | 55.23 | 55.56 | 65.61 | 60.46 | 32.80 | 50.92 | 76.31 |
| | QuaRot-RTN | 36.26 | 58.38 | 53.98 | 56.91 | 67.08 | 72.78 | 30.80 | 53.74 | 79.42 |
| | GPTQ | 38.91 | 56.90 | 60.26 | 60.54 | 68.44 | 65.38 | 35.20 | 55.09 | 82.49 |
| | QuaRot | 42.06 | 64.69 | 61.78 | 62.43 | 70.02 | 73.39 | 33.80 | 58.31 | 86.62 |
| | SpinQuant | 41.89 | 60.61 | 59.42 | 59.27 | 68.88 | 68.96 | 35.00 | 56.29 | 84.14 |
| | OSTQuant | 43.86 | 67.68 | 64.85 | 61.72 | 71.27 | 80.86 | 37.20 | 61.06 | 90.82 |
| | FlatQuant† | 42.32 | 71.38 | 63.76 | 61.64 | 71.22 | 81.41 | 38.80 | 61.50 | 91.50 |
| | MR-GPTQ | 44.45 | 68.56 | 64.27 | 63.61 | 71.76 | 76.64 | 37.80 | 61.01 | 91.02 |
| | **LATMiX-LU (Ours)** | 46.42 | 70.45 | 66.36 | 63.93 | 73.39 | 78.44 | 38.80 | 63.63 | 94.89 |
| | **LATMiX-QR (Ours)** | 47.27 | 72.60 | 65.10 | 62.19 | 72.69 | 80.95 | 38.40 | **63.70** | **94.73** |

*Table 21.* Performance of Qwen3-8B across weight and activation quantization configurations.

| Format | Method | ARC-C | ARC-E | HellaSwag | WinoGrande | PIQA | BOOLQ | OBQA | Avg. Accuracy | Avg. Recovery (%) |
|---|---|---|---|---|---|---|---|---|---|---|
| | FP16 | 56.74 | 80.68 | 76.53 | 70.17 | 77.69 | 86.64 | 41.60 | 70.01 | 100 |
| MXFP4 | RTN | 48.04 | 73.95 | 70.01 | 64.96 | 73.83 | 83.03 | 39.20 | 64.72 | 92.21 |
| | QuaRot-RTN | 48.12 | 71.38 | 66.91 | 65.67 | 72.80 | 82.42 | 36.00 | 63.33 | 89.95 |
| | GPTQ | 48.98 | 74.54 | 70.15 | 64.64 | 74.16 | 83.21 | 39.60 | 65.04 | 92.74 |
| | QuaRot | 48.38 | 72.98 | 71.18 | 67.96 | 73.78 | 85.14 | 38.80 | 65.46 | 93.15 |
| | SpinQuant | 50.51 | 74.87 | 71.17 | 67.64 | 74.81 | 84.13 | 40.40 | 66.22 | 94.53 |
| | OSTQuant | 52.30 | 76.89 | 72.04 | 67.48 | 76.16 | 85.32 | 40.80 | 67.28 | 96.05 |
| | FlatQuant† | 52.13 | 76.09 | 72.55 | 68.03 | 75.41 | 84.62 | 39.40 | 66.89 | 95.34 |
| | MR-GPTQ | 52.13 | 78.37 | 72.38 | 67.40 | 75.95 | 84.95 | 42.80 | 67.71 | 96.91 |
| | **LATMiX-LU (Ours)** | 54.69 | 77.57 | 74.06 | 69.93 | 76.33 | 85.81 | 42.20 | **67.94** | **97.07** |
| | **LATMiX-QR (Ours)** | 54.09 | 78.45 | 73.84 | 70.87 | 77.31 | 85.10 | 40.80 | 68.64 | 97.97 |
| MXINT4 | RTN | 43.34 | 68.14 | 65.52 | 59.59 | 73.07 | 74.71 | 35.40 | 59.97 | 85.25 |
| | QuaRot-RTN | 41.21 | 64.27 | 64.64 | 62.75 | 71.16 | 75.60 | 35.00 | 59.23 | 84.17 |
| | GPTQ | 45.48 | 66.04 | 66.16 | 62.90 | 73.50 | 80.12 | 39.00 | 61.89 | 88.42 |
| | QuaRot | 46.59 | 68.31 | 70.67 | 65.75 | 74.10 | 82.39 | 40.00 | 63.97 | 91.35 |
| | SpinQuant | 46.33 | 71.68 | 69.54 | 65.67 | 73.94 | 81.83 | 36.80 | 63.68 | 90.43 |
| | OSTQuant | 52.99 | 75.55 | 72.08 | 67.48 | 75.79 | 84.25 | 41.00 | 67.02 | 95.82 |
| | FlatQuant† | 48.55 | 74.92 | 70.98 | 67.01 | 75.57 | 83.76 | 39.80 | 65.80 | 93.76 |
| | MR-GPTQ | 51.88 | 76.09 | 72.10 | 68.27 | 75.68 | 83.73 | 39.00 | 66.68 | 95.01 |
| | **LATMiX-LU (Ours)** | 50.09 | 75.88 | 74.12 | 67.17 | 75.84 | 84.40 | 41.80 | **68.10** | **97.16** |
| | **LATMiX-QR (Ours)** | 52.73 | 76.01 | 71.23 | 67.25 | 76.39 | 83.24 | 39.60 | 67.09 | 96.00 |

*Table 22.* Performance of Qwen3-14B across weight and activation quantization configurations.

| Format | Method | ARC-C | ARC-E | HellaSwag | WinoGrande | PIQA | BOOLQ | OBQA | Avg. Accuracy | Avg. Recovery (%) |
|---|---|---|---|---|---|---|---|---|---|---|
| | FP16 | 60.23 | 82.82 | 79.86 | 74.58 | 79.86 | 89.32 | 46.40 | 73.29 | 100 |
| MXFP4 | RTN | 55.88 | 78.36 | 75.84 | 71.90 | 78.01 | 86.45 | 42.20 | 69.80 | 94.88 |
| | QuaRot-RTN | 54.35 | 76.01 | 73.25 | 70.08 | 76.82 | 85.77 | 41.80 | 68.30 | 92.85 |
| | GPTQ | 58.61 | 79.20 | 75.86 | 72.13 | 78.23 | 87.18 | 43.00 | 70.60 | 96.12 |
| | QuaRot | 56.82 | 80.38 | 76.19 | 74.34 | 78.07 | 88.50 | 45.00 | 71.33 | 97.18 |
| | SpinQuant | 57.25 | 80.00 | 76.79 | 72.13 | 78.61 | 87.88 | 44.00 | 70.95 | 96.59 |
| | OSTQuant | 58.95 | 81.01 | 77.91 | 73.40 | 79.32 | 87.82 | 43.80 | 71.74 | 97.67 |
| | FlatQuant† | 55.80 | 79.41 | 76.47 | 73.48 | 77.04 | 87.55 | 44.20 | 70.56 | 96.07 |
| | MR-GPTQ | 58.10 | 81.14 | 77.31 | 71.90 | 78.29 | 86.57 | 46.00 | 71.33 | 97.39 |
| | **LATMiX-LU (Ours)** | 59.30 | 81.77 | 77.81 | 71.66 | 79.16 | 88.25 | 44.80 | **71.82** | **97.99** |
| | **LATMiX-QR (Ours)** | 56.82 | 80.93 | 77.22 | 72.77 | 78.34 | 88.65 | 44.80 | 71.36 | 97.36 |
| MXINT4 | RTN | 53.15 | 74.53 | 72.51 | 67.08 | 76.98 | 82.20 | 41.20 | 66.81 | 90.88 |
| | QuaRot-RTN | 49.06 | 72.09 | 72.00 | 68.19 | 75.84 | 84.92 | 38.20 | 65.76 | 88.92 |
| | GPTQ | 53.49 | 77.52 | 72.89 | 68.82 | 76.65 | 83.02 | 40.80 | 67.59 | 91.83 |
| | QuaRot | 57.42 | 82.28 | 76.05 | 71.03 | 77.91 | 87.52 | 43.00 | 70.74 | 96.19 |
| | SpinQuant | 56.48 | 79.96 | 74.83 | 70.79 | 78.23 | 87.15 | 43.30 | 70.11 | 95.39 |
| | OSTQuant | 58.36 | 81.14 | 77.74 | 72.13 | 78.34 | 86.63 | 44.00 | 71.19 | 96.97 |
| | FlatQuant† | 55.80 | 76.72 | 74.74 | 70.63 | 75.89 | 85.59 | 44.60 | 69.14 | 94.36 |
| | MR-GPTQ | 58.61 | 80.30 | 76.56 | 72.05 | 78.99 | 87.76 | 42.60 | 70.98 | 96.53 |
| | **LATMiX-LU (Ours)** | 58.36 | 80.80 | 78.21 | 72.05 | 79.59 | 87.52 | 46.00 | **71.79** | **97.95** |
| | **LATMiX-QR (Ours)** | 57.25 | 81.27 | 75.43 | 71.03 | 77.42 | 86.51 | 43.80 | 70.39 | 96.04 |

*Table 23.* Performance of Qwen3-32B across weight and activation quantization configurations.

| Format | Method | ARC-C | ARC-E | HellaSwag | WinoGrande | PIQA | BOOLQ | OBQA | Avg. Accuracy | Avg. Recovery (%) |
|---|---|---|---|---|---|---|---|---|---|---|
| | FP16 | 60.92 | 83.29 | 83.95 | 76.87 | 82.15 | 86.45 | 46.40 | 74.29 | 100 |
| MXFP4 | RTN | 57.08 | 78.45 | 79.42 | 70.56 | 78.50 | 85.53 | 46.80 | 70.91 | 95.66 |
| | QuaRot-RTN | 58.70 | 76.64 | 77.96 | 72.77 | 77.63 | 82.93 | 42.60 | 69.89 | 94.01 |
| | GPTQ | 58.27 | 81.14 | 80.78 | 74.98 | 79.97 | 82.84 | 45.80 | 71.40 | 96.95 |
| | QuaRot | 56.99 | 80.13 | 81.73 | 74.19 | 79.37 | 87.00 | 46.80 | 72.32 | 97.39 |
| | SpinQuant | 59.64 | 78.45 | 80.24 | 72.05 | 77.63 | 82.66 | 45.20 | 70.83 | 95.56 |
| | OSTQuant | 60.66 | 82.99 | 81.19 | 74.11 | 79.32 | 87.06 | 47.40 | 73.25 | 98.82 |
| | FlatQuant[†] | 57.42 | 79.04 | 80.76 | 74.66 | 78.56 | 85.84 | 42.60 | 71.27 | 95.60 |
| | MR-GPTQ | 60.40 | 80.51 | 81.95 | 75.45 | 79.43 | 84.58 | 46.40 | 72.67 | 98.01 |
| | **LATMiX-LU (Ours)** | 60.49 | 81.77 | 82.04 | 76.32 | 79.86 | 86.39 | 47.20 | **73.44** | **99.04** |
| | **LATMiX-QR (Ours)** | 59.81 | 81.22 | 80.20 | 74.03 | 78.61 | 87.67 | 43.80 | 72.19 | 97.01 |
| MXINT4 | RTN | 50.59 | 69.14 | 66.60 | 63.14 | 73.55 | 80.09 | 40.20 | 63.33 | 85.19 |
| | QuaRot-RTN | 55.63 | 77.10 | 74.66 | 69.37 | 76.60 | 83.85 | 41.80 | 68.43 | 91.91 |
| | GPTQ | 55.54 | 77.35 | 78.26 | 70.79 | 78.81 | 85.16 | 44.20 | 70.02 | 94.15 |
| | QuaRot | 60.15 | 79.92 | 81.21 | 73.40 | 79.70 | 84.49 | 44.00 | 71.84 | 96.64 |
| | SpinQuant | 55.46 | 76.43 | 79.51 | 71.03 | 78.12 | 84.67 | 44.60 | 69.97 | 94.15 |
| | OSTQuant | 60.75 | 82.07 | 81.11 | 75.76 | 78.89 | 87.67 | 45.20 | 73.06 | 98.32 |
| | FlatQuant[†] | 56.74 | 76.93 | 78.86 | 74.11 | 77.80 | 85.81 | 44.40 | 70.66 | 95.07 |
| | MR-GPTQ | 58.70 | 80.93 | 81.65 | 73.95 | 78.99 | 84.89 | 47.00 | 72.30 | 97.51 |
| | **LATMiX-LU (Ours)** | 62.62 | 82.36 | 81.57 | 76.32 | 79.86 | 88.16 | 46.00 | **73.84** | **99.49** |
| | **LATMiX-QR (Ours)** | 56.82 | 80.51 | 79.47 | 73.16 | 79.10 | 86.94 | 44.80 | 71.54 | 96.16 |

