# OpenReview forum: "LATMiX: Learnable Affine Transformations for Microscaling Quantization of LLMs"
_ICML.cc/2026/Conference — ICML 2026 regular_

### Official Review · Reviewer_bwAi · 2026-03-04

**Soundness:** 2
**Presentation:** 3
**Significance:** 2
**Originality:** 3
**Overall Recommendation:** 4
**Confidence:** 4

**Summary:**

The paper proposes a PTQ method called LATMiX for the MX format by replacing the learnable rotation transformation in SpinQuant with learnable affine transformations. The paper provides a detailed description of the parametrization of invertible affine transformations and explains the advantages of the proposed transformation compared with those used in previous methods. The paper further empirically demonstrates that LATMiX outperforms existing PTQ methods on MXFP4 for models with sizes below 8B parameters.

**Compliance With Llm Reviewing Policy:**

Affirmed.

**Final Justification:**

The authors have adequately addressed my primary concerns by providing comparisons with FlatQuant and with rotation methods that include bias, thereby improving the overall soundness of the work. Accordingly, I have raised my score to weak accept.

**Key Questions For Authors:**

See weaknesses.

**Limitations:**

No. See weaknesses.

**Strengths And Weaknesses:**

**Strengths:**
1. The paper is well written and easy to follow.
2. The paper comprehensively demonstrates the effectiveness of the proposed learnable affine transformations on MXFP4 and clearly shows their advantages over previously used transformations.
3. The proposed LATMiX-LU and LATMiX-QR methods enable effective training of invertible affine transformations.

**Weaknesses:**
1. As described in Appendix D.2, the learnable matrix $A_1$ is changed from being designed for each individual weight matrix to a global transformation. This modification removes one of the key advantages of FlatQuant. However, Section 5.1 Experimental Settings and Table 1 do not provide any clarification regarding this change. In addition, after switching to a global transformation, it is unclear how $A_1$ is updated when FlatQuant performs training at the level of individual transformer blocks.
2. The paper lacks experimental comparisons on models larger than 8B parameters. Larger models have stronger practical demand for extreme quantization.
3. The paper does not include comparisons with similar methods such as OSTQuant [1].
4. The transformation ablation study does not include the configuration using learnable rotation transformation with bias.
5. The paper does not report comparisons for the case where only $T_1$ and $T_2$ are fused without quantization with the full-precision baseline. Since the proposed transformation does not preserve computational invariance, the impact introduced by the transformations themselves, independent of quantization, should be explicitly reported.
6. Figure 4 should be moved to the main body of the paper to facilitate understanding of the overall architecture. The results in Appendix E.2 Initialization of the Transformations should also be included in the main text to clearly present the choice of the initialization strategy, which is a critical component of methods using learnable transformations.
7. Figure 2(b) appears to be missing the result for the vanilla baseline.

[1] Hu, Xing, et al. "OSTQuant: Refining Large Language Model Quantization with Orthogonal and Scaling Transformations for Better Distribution Fitting." The Thirteenth International Conference on Learning Representations.

---

> ### Author Rebuttal · Authors · 2026-03-31
>
> We thank the reviewer for the constructive and insightful feedback. We are encouraged that the reviewer appreciated the effectiveness of LATMiX, the experimental evaluations, and our proposed approaches. Missing references, relevant discussions, and clarifications will be added to the next version of the paper.
> - **Large models:** We first would like to point out that our focus was mainly on models that can fit on the edge, hence we put more emphasis on smaller models. Furthermore, as the models get bigger, the performance gap between GPTQ and FP16 gets smaller, as seen in (Egiazarian et al., 2025). Nevertheless, following the reviewer's suggestion, we have conducted additional experiments with the larger models Qwen3-14B and Qwen3-32B, and we report the results in the Tables below. From the tables, LATMiX outperforms all baselines, reaching near FP performance in Qwen3-32B. Due to time constraints and limited resources, we were unable to evaluate LATMiX and baseline methods on larger models, yet we expect the performance gap to persist.
>
> **Qwen3-14B**
> |Format | Method | Acc. | Recovery
> |----|---|---|---
> |  | FP16 | 73.29 | 100 |
> | MXFP4 | SpinQuant | 70.95 | 96.80
> | | FlatQuant| 70.56 | 96.27
> |  | MR-GPTQ | 71.33 | 97.32
> |  | **LATMiX** | **71.82** | **97.99**
> |----|---|---|---
> | MXINT4 | SpinQuant | 70.11 | 95.66
> | | FlatQuant| 69.14 | 94.33
> |  | MR-GPTQ | 70.98 | 96.84
> |  | **LATMiX** | **71.79** | **97.95**
>
> **Qwen3-32B**
> |Format | Method | Acc. | Recovery
> |----|---|---|---
> |  | FP16 | 74.29 | 100 |
> | MXFP4 | GPTQ | 71.40 | 96.10 |
> |  | FlatQuant | 71.27 | 95.93 |
> |  | MR-GPTQ | 72.67 | 97.81 |
> |  | **LATMiX** | **73.44** | **98.85** |
> |----|---|---|---
> | MXINT4 | GPTQ | 70.02 | 94.25 |
> |  | FlatQuant | 70.66 | 95.11 |
> |  | MR-GPTQ | 72.30 | 97.32 |
> |  | **LATMiX** | **73.84** | **99.39** |
>
> - **Comparison to OSTQuant:** In the paper we compared LATMiX to baselines designed specific for MX data format, such as MR-GPTQ, as well as other recent strong non-MX baselines, such as QuaRot, SpinQuant, and FlatQuant. Following the reviewer comment, we have now added a comparison to OSTQuant as well in the tables below. From the tables, it can be seen that LATMiX outperforms this baseline across all models and tasks.
>
> **PPL**
> | Method | Llama-1B | Qwen-1.7B | Llama-8B |
> |---|---:|---:|---:|
> | OSTQuant | 13.3 | 18.35 | 8.88 |
> | LATMiX | **11.74** | **17.73** | **7.97** |
>
> **Avg 0-Shot Accuracy**
> | Method | Llama-1B | Qwen-1.7B | Llama-8B |
> |---|---:|---:|---:|
> | OSTQuant | 52.57 | 55.48 | 67.02 |
> | LATMiX | **54.18** | **56.91** | **68.46** |
>
> - **Effect of transforms without quantization:** We thank the reviewer for this important suggestion. LATMiX optimizes a distillation loss so that the transformed quantized model remains approximately consistent with the float model, rather than exactly invariant. To validate that, we evaluated the Qwen3-8B full-precision model after fusing the learned $T_1$ and $T_2$, without any quantization, at multiple training steps. Compared to the FP16 baseline, the transformed models show negligible changes in both WikiText2 perplexity and PIQA accuracy. This indicates that the learned transforms have minimal effect on the float model.
>
> | Training steps | PPL | PIQA |
> |---:|---:|---:|
> | FP16 | 10.246 | 77.69 |
> | 0 | 10.243 | 77.69 |
> | 1 | 10.242 | 77.58 |
> | 100 | 10.240 | 77.64 |
> | 200 | 10.253 | 77.53 |
> | 500 | 10.248 | 77.42 |
> | 1000 | 10.236 | 77.48 |
>
> - **FlatQuant’s $A_1$ matrix update:** As FlatQuant learns a matrix per each block, a direct and fair comparison is infeasible. Hence, we attempted to stay loyal to the FlatQuant implementation while keeping the setup similar to ours, which is the common setup in the literature. The matrix $A_1$ is parametrized as described in Section 3.1 in FlatQuant’s paper, i.e., taking the Kronecker product of two lightweight matrices, both of which are learned in the optimization process. This baseline further allowed us to measure the quality of the affine parameterizations proposed in LATMiX, showing favorable results to our proposed approach.
>
> - **Rotation with bias:**  Please see the results in the table below. Adding a bias improves upon the orthogonal matrix, but lacks the additional flexibility of LATMiX.
>
> | Method | Llama-3.2-1B | Qwen3-1.7B |
> |---|---:|---:|
> | Orth. Matrix | 13.35 | 17.7 |
> | **Orth. Matrix + bias** | 11.99 | 17.28 |
> | LATMiX | **11.69** | **16.98** |
>
> - **Moving details to the main text:** We will modify the main text to account for these details under the page limitation constraints in the next version of the paper.
> - **Missing case in Fig. 2(b):** We thank the reviewer for highlighting this oversight. We added the Vanilla case result to the figure which can be seen in this anonymous
> [[LINK]](https://anonymous.4open.science/r/latmix-rebuttal-1/ppl_per_block_error_revised.png).

---

> > ### Author Rebuttal · Reviewer_bwAi · 2026-04-02
> >
> > Thank you for your response. Your answers address some of my concerns, but important issues still remain. Therefore, I maintain my score.
> >
> > In the comparison with FlatQuant, the original trade-off in FlatQuant should not be removed, since doing so eliminates one of the key advantages that FlatQuant derives from that trade-off. The fusion strategy requires $A_1$ to be identical across all Transformer blocks. By contrast, FlatQuant uses the Kronecker product of two small matrices in order to allow different transformations for different blocks without introducing excessive overhead.

---

> > > ### Author Response · Authors · 2026-04-03
> > >
> > > We thank the reviewer for the response and the engagement in the rebuttal of our paper. We are glad that we addressed some of the reviewer's concerns, and we will be happy to address any additional concerns in case there are ones. The reviewer’s feedback is highly valuable in shaping and making our paper better.
> > >
> > > Just to reiterate our initial response, our motivation in the implementation of FlatQuant was two-fold, first we aimed to align the setup of all methods to have exactly the same transformation pipeline. Second, it served as an additional method for constructing affine transformations under exactly the same setup. While we believe that the proposed comparison in the paper is valuable, we also accept and appreciate the reviewer’s stance. Hence, we now add additional results on zero-shot accuracy in the table below comparing LATMiX to FlatQuant in its original formulation (marked in bold), namely having an invertible transformation for each transformer block with the associated loss function.
> > > We report here initial results; a full comparison will be added in the next version of the paper.
> > >
> > > | Method | Llama-1B | Llama-8B | Qwen-1.7B
> > >  |---|---:|---:|---:|
> > >  | FlatQuant’s matrix structure | 52.45| 67.81| 54.85 |
> > >  | **FlatQuant** | 52.96 | 67.93 | 55.35 |
> > >  | LATMiX | **54.18** | **68.46** | **56.91** |
> > >
> > > We indeed observe that introducing per-block transformations improves FlatQuant’s performance. Nevertheless, LATMiX consistently outperforms FlatQuant. We hypothesize that this is because FlatQuant was originally designed for integer quantization without block-wise scaling, whereas our setting involves microscaling. This mismatch may limit the effectiveness of its online transformation in this regime.
> > >
> > > More broadly, this suggests an interesting open direction: designing online transformations that are specifically tailored for microscaling quantization. We will include these additional results and expand the discussion on online transformations in the revised version of the paper. We hope that our comment with the new results and our answers to all of the reviewers initial comments eliminate any concerns the reviewer might have.

---

### Official Review · Reviewer_tUsN · 2026-03-09

**Soundness:** 3
**Presentation:** 3
**Significance:** 3
**Originality:** 3
**Overall Recommendation:** 4
**Confidence:** 4

**Summary:**

This paper studies post-training quantization of LLMs under microscaling (MX) formats (e.g., MXFP4 / MXINT4), focusing on the combination of MX’s block-wise scaling and global activation transformations. The authors provide a theoretical bound on the MX quantization MSE under general affine transforms, arguing that good transformations must account for both the MX block structure and activation statistics. Building on this, the authors propose LATMiX, which learns invertible affine transformations trained with a KL distillation loss plus a volume-preserving regularizer to maintain stable invertibility. The authors then fold the resulting transformations into adjacent linear layers to avoid (or minimize) inference overhead. Experiments across multiple Llama3/Qwen3 sizes report consistent improvements over several MX baselines and transformation-based PTQ methods on zero-shot benchmarks and WikiText2 perplexity.

**Compliance With Llm Reviewing Policy:**

Affirmed.

**Final Justification:**

The rebuttal addressed some of my concerns and reinforced my prior assessment. I am still inclined to recommend this paper for acceptance.

**Key Questions For Authors:**

- Why does “block index” matter in Fig. 2c? Are blocks shown in original channel order, or sorted by some statistic? When you say “dominant (early-index) blocks,” what determines that dominance?
- Can you summarize (in main text or a table) sensitivity to λ (volume regularizer), distillation temperature, number of steps, and calibration set size, especially for downstream zero-shot accuracy (not only WikiText2 perplexity)?
- You mention LU often slightly wins and QR may be harder due to the matrix exponential. Can you report training time, stability (e.g., condition number / min singular value trends), and when QR is preferable (if  ever)?
- For models without a bias in the linear layer, what is the inference overhead of the affine shift?
- My intuition is that online transformations might have an advantage over merged/folded transformations, since the transformation itself is not constrained by the quantized weights and may retain higher effective precision. However, your results show LATMiX consistently outperforming FlatQuant, which relies on online transformations. Do you have intuition for why the merged approach performs better here?
- (Optional) Anonymous access to the code would make it easier to validate the folding details and baseline re-implementations, although I understand this may not always be possible during review.

**Limitations:**

Limitations are mentioned but see weakness for which are inadequately addressed (e.g., calibration overhead).

**Strengths And Weaknesses:**

**Strengths:**

1. The paper directly targets the intersection of MX block-wise scaling and global transformations, and offers both an analysis and a practical method tailored to this setting.
2. While not the first MX quantization paper overall, it is among the more focused attempts to design/learn activation transformations specifically for MX.
3. The LU/QR parameterizations provide a pragmatic way to optimize general invertible matrices, avoiding  constrained optimization on the Steifel manifold.
4. Extending from linear transforms to affine transforms is a meaningful generalization; the paper motivates this via the bound and notes the shift can often be folded into existing biases (or implemented with minimal overhead).
5. Using KL distillation to relax exact functional equivalence (especially given non-orthogonal transforms) plus a volume regularizer is a sensible and empirically motivated combination.
6. For the most part, the paper is well written, and I appreciated the theoretical grounding for the approach (particularly Theorem 3.3 and the following remarks).

**Weaknesses:**

1. The use of distillation/KL for quantization is well-established (e.g., [1,2]); the paper’s novelty is more in applying it to learn foldable affine transforms under MX, but the related-work positioning with respect to distillation-based PTQ/QAT could be sharper.
2. Beyond the paper’s own loss ablations, it would strengthen the story to (i) isolate the impact of KL loss for SpinQuant as has been done for FlatQuant, and (ii) show sensitivity to training choices (λ, temperature, steps, calibration size) on downstream zero-shot metrics.
3. The paper claims to incorporate both MX block structure and feature distribution, but the mechanistic distinction could be explained more explicitly in the main text. What prevents the same recipe from being applied to prior methods under MX? How does this perform non-MX datatypes?
4. Equation (1) presentation is hard to read, and Eq. (3) switches indexing conventions (e.g., blocks starting at 0 for Eq. 1). Could you restate near Theorem 3.3 and/or make indexing consistent across Sec. 2–3?
5. The LATMiX text styling in Fig. 1 looks inconsistent with the rest of the figure; tightening visual consistency would improve polish.
6. Non-orthogonal affine transforms break RMSNorm invariance; the paper’s approach is plausible, but the main text should more directly explain why this remains stable and whether there are failure modes.
7. LATMiX requires an optimization stage (e.g., ~1k–2.5k steps depending on parameterization) and teacher/student forward passes; beyond what is noted in the conclusion, a concrete wall-clock/memory overhead summary would help readers judge practicality.
8. Lines 153-157 are ambiguous. It initially reads as if to say there is severe degradation with MR-GPTQ and BRQ. Alternately, it could mean that MX + block Hadamard rotations + round-to-nearest (RTN) experiences degradation. Either way, the data does not support such a strong claim. This comment should be made more precise.
9. Appendix D.2 explicitly states baseline deviations (e.g., FlatQuant evaluated with the authors’ distillation loss for the reported main results, and SpinQuant omits an online Hadamard on RoPE outputs). There should be (a) clearer reporting of these deviations in the main text, and/or (b) an “as-original” baseline run where feasible.

[1]  Liu et al. "LLM-QAT: Data-free quantization aware training for large language models." ACL 2024

[2]  Hu et al. "OSTQuant: Refining large language model quantization with orthogonal and scaling transformations for better distribution fitting." ICLR 2025.

---

> ### Author Rebuttal · Authors · 2026-03-31
>
> We thank the reviewer for the constructive and insightful feedback. We are delighted that the reviewer appreciated the setup in which our paper operates, the theory motivating LATMiX, and our practical algorithm. Missing references, ablations, clarifications and relevant discussions will be incorporated in the revised version of the paper. Additional results can be found in this anonymous [[LINK]](https://anonymous.4open.science/r/latmix-rebuttal-1).
> - **Distillation/KL loss:** Thank you for the references. Please see a comparison to [2] in our comments to Reviewer bwAi, showing favourable performance for LATMiX over OSTQuant.
> - **Mechanistic distinction between MX block structure and feature distribution:** We will be happy for clarifications in case we misunderstood the questions. The MX block structure accounts for both the number of blocks and the scaling, while the feature distribution is the distribution of the activation features at some NN layer. Theorem 3.3 shows that both affect the overall quantization error. Prior methods can be applied to MX, but if they do not account for both (e.g., suitable transformation and initialization), they may suffer from large errors. We believe that LATMiX can be very effective in non-MX setups as well (corresponding to $B=d$), but we left this investigation for future work.
> - **Calibration set size and runtime; inference overhead:** Please see our response to Reviewer Mb4i for the wall-clock training time (~2 hours; comparable with baselines), our response to Reviewer Tfqu for the robustness of LATMiX to the calibration set size, and our response to Reviewer Tfqu for the throughput measurements (LATMiX is comparable to baseline methods during inference).
> - **RMSNorm invariance:** Please see our response to reviewer Tfqu that touches upon this point.
> - **Degradation claim ambiguity:**  The intended meaning was to refer to prior works (MR-GPT, BRQ), which report that combining MX quantization with full rotations can degrade outlier suppression and lead to performance drops. This observation is also consistent with our empirical results in Table 1. We will revise the text accordingly.
> - **Deviations from SpinQuant and FlatQuant setup:** To ensure a fair comparison we used exactly the same setup for all methods. Regarding FlatQuant, please see our response to Reviewer bwAi, where this point is addressed in more detail. Regarding SpinQuant, as we followed the setup used by MR-GPTQ we did not use online transformations. Adding these transformations should benefit all methods. Furthermore, to eliminate bias due to loss function we tested both SpinQuant and FlatQuant with LATMiX distillation loss as well as the proposed loss function in each approach, and took the best one. For SpinQuant it was the cross-entropy loss. See below Wikitext2 PPL results for SpinQuant with both losses:
> | Loss Function | Llama-1B | Qwen-1.7B | Qwen-8B | Llama-8B |
> |---|---:|---:|---:|---:|
> | LATMiX Loss | 13.2 | 26.22 | 11.16 | 8.89 |
> | CE| 12.74 | 19.21 | 9.55 | 8.84 |
>
> - **Block reference in Fig. 2c:** In Fig. 2c, block indices correspond to the model’s original transformer block order and are not sorted by any statistic. In the specific example shown, some of the earlier blocks have larger error values by chance. Dominant blocks are ones with the highest block-wise quantization error compared to the vanilla (no transformation) case.
> - **Ablations:** We add here the proposed ablation studies. Overall LATMiX is robust to many choices, such as the number of training steps and calibration set size. Setup used in the paper is denoted in **bold** (additional ablations are in Tables 2-5 at the anonymous link)
>
> *Num. of Training steps (Llama-8B)*
> |  | PPL | Avg. 0-Shot |
> |---|---|---|
> | 100 | 8.07 | 67.48 |
> | 500 | 8.01 | 68.55 |
> | **1000** | 7.97 | 68.46 |
> | 5000 | 8.02 | 68.94 |
>
> *Calibration set size*
> |  | Llama-8B | Qwen-8B |
> |---|---|---|
> | 1 | 69.11 | 66.33 |
> | 64 | 70.32 | 68.16 |
> | 128 | 70.31 | 68.12 |
> |**256** | 70.51 | 68.46 |
> | 512 | 71.99 | 69.68 |
>
> *Lambda (Qwen-1.7B)*
> |  | PPL | Avg 0-Shot |
> |---:|---:|---:|
> | 0.01 | 17.67 | 56.23 |
> | **0.1** | 17.37 | 56.91 |
> | 0.5 | 17.4 | 56.15 |
> | 1 | 18.18 | 56.41 |
>
> *Temperature (Llama-8B)*
> |  | PPL | Avg 0-Shot |
> |---:|---:|---:|
> | 0.5 | 8.47 | 68.89 |
> | 1 | 8.04 | 68.37 |
> | **1.5** | 7.97 | 68.46 |
> | 5 | 8.08 | 67.87 |
> - **LU vs. QR representation:** LU and QR training time is roughly the same. Please see at the anonymous link a figure tracking the condition number along training; for both it is stable. The QR spans a wider class of transformations but perhaps it is harder to optimize due to matrix exponential. We believe that careful optimization of $G$ will help it to get better results.
> - **Code for review:** A file with LATMiX implementation was submitted under supplementary materials in the original submission. An optimized and more comprehensive version of the code will be provided with the revised submission of the paper.

---

> > ### Author Rebuttal · Reviewer_tUsN · 2026-04-02
> >
> > Thank you for your responses. I will keep my score.
> >
> > I would still like to understand any intuition you may have captured on the online vs. folded transformations (see Question 5).

---

> > > ### Author Response · Authors · 2026-04-03
> > >
> > > We thank the reviewer for the engagement and highlighting this important point. The reviewer’s feedback is highly valuable in shaping and making our paper better.
> > >
> > > We agree that online transformations have the potential to achieve higher accuracy compared to merged (offline/folded) transformations. However, this may come with additional inference latency and the need for specialized kernel implementations. These practical considerations motivated our focus on folded setups, such as SpinQuant, OSTQuant and QuaRoT. Furthermore, there may be an implicit bias effect stemming from sharing the parameters of $T_1$, helping to combat overfitting.
> > >
> > > Regarding the comparison with FlatQuant, the original results evaluated only the transformation structure of FlatQuant, rather than compression to its full pipeline, which is important for comparing different transformation representations. Following the feedback from Reviewer bwAi, we extended the comparison to include the fully compressed setting for FlatQuant, incorporating per-block (online) transformations and the associated loss terms (marked in bold at the table). Results on zero-shot accuracy are reported in the table below.
> > > We report here initial results; a full comparison will be added in the next version of the paper.
> > >
> > >  | Method | Llama-1B | Llama-8B | Qwen-1.7B
> > >  |---|---:|---:|---:|
> > >  | FlatQuant’s matrix structure | 52.45| 67.81| 54.85 |
> > >  | **FlatQuant** | 52.96 | 67.93 | 55.35 |
> > >  | LATMiX | **54.18** | **68.46** | **56.91** |
> > >
> > > We indeed observe that introducing per-block transformations improves FlatQuant’s performance. Nevertheless, LATMiX consistently outperforms FlatQuant. We hypothesize that this is because FlatQuant was originally designed for integer quantization without block-wise scaling, whereas our setting involves microscaling. This mismatch may limit the effectiveness of its online transformation in this regime.
> > >
> > > More broadly, this suggests an interesting open direction: designing online transformations that are specifically tailored for microscaling quantization. We will include these additional results and expand the discussion on online transformations in the revised version of the paper.

---

### Official Review · Reviewer_Tfqu · 2026-03-10

**Soundness:** 3
**Presentation:** 3
**Significance:** 2
**Originality:** 2
**Overall Recommendation:** 4
**Confidence:** 4

**Summary:**

This paper introduces LATMiX, a learnable affine-transformation–based post‑training quantization (PTQ) approach designed for microscaling formats. LATMiX jointly learns affine transformations using distillation and spectral normalization, with the goal of balancing intra‑block and inter‑block quantization error. Empirically, the method is shown to outperform global rotations and deliver improvements over recent quantization baselines across a broad range of LLMs and tasks.

**Compliance With Llm Reviewing Policy:**

Affirmed.

**Final Justification:**

The authors addressed some of my main issues but some crucial aspects (definition of $T_3$, and assessment of effect of violating computational invariance) remain unclear. However, I think the authors' response contributed to strengthening the submission solving some of my main concerns.

UPDATE: the authors solved my main concern during the second phase of the rebuttal so I updated my score.

**Key Questions For Authors:**

1. **Calibration Overfitting:**
Given that LATMiX introduces non‑orthogonal transformations and breaks equivalence with the base model, how does the method avoid overfitting to the specific calibration set? Have the authors tested sensitivity to different calibration distributions?

2. **Computational Overhead:**
Could the authors provide a more detailed analysis or measurements supporting the claim that the learned transformations incur “no additional inference cost,” especially considering $T_3$  is not fused and involves dense matrix multiplications?

3. **Validity of the Local Error Metric:**
Since local quantization error does not strongly predict downstream accuracy (Figure 2a–b), how do the authors justify using it as the central theoretical objective? What are its limitations?

4. **Inconsistency in Hadamard Analysis:**
Which results support the claim that full Hadamard slightly outperforms block‑Hadamard for small block sizes? This seems inconsistent with Figure 2. Can the authors clarify the empirical setup?

5. **Applicability of Sub‑Gaussian Assumptions:**
Given the heavy‑tailed distributions reported for activations, how does Equation 4 remain relevant? Are there guarantees or heuristics supporting its use under super‑Gaussian distributions?

6. **Lack of single transform ablation:**
How much does each separate transform impact the model performance? In particular, how much effect does the non-orthogonal component of $T_1$ have? Why is $T_3$ not fused into the weights? How does the model perform with $T_3$ fused into the weights?

**Limitations:**

The paper does does not mention specific limitation with the exception of the need for training data. But the extra compute due to the the transform choice and the extra risk of overfitting due to the violation of computational equivalence are not addressed

**Strengths And Weaknesses:**

# Strengths
* The paper proposes a simple yet effective method, motivated by clear limitations of existing rotation‑based PTQ techniques.

* The empirical evaluation is comprehensive, covering multiple LLMs and benchmark tasks. The improvements reported are consistent and measurable across datasets and model sizes.

# Weaknesses


 ## Primary concerns


1. **Computational cost and fusion of transformations:**
While the text claims that the learned transformations can be folded into linear layers “with no additional inference cost,” this is difficult to reconcile with Figure 4 showing that $T_3$ is not fused (and is the largest, most expensive transform). Furthermore, since arbitrary dense matrices cannot leverage fast algorithms such as the Fast Hadamard Transform, the cost of applying $T_3$ even once may be considerably higher than structured rotations. A more rigorous justification of the computational‑cost claim is needed.



2. **Deviation from computational equivalence:**
Section 3.2 notes that since $T_1$ contains non-orthogonal components, $T_1^{-1}RMSNorm(T_1 x) \neq  RMSNorm(x)$, meaning the model after applying LATMiX is no longer computationally equivalent to the original model. This introduces additional degrees of freedom, effectively altering the network’s behavior. However, the paper does not study how sensitive the method is to the choice of calibration data, nor whether the additional degrees of freedom allow the model to inadvertently overfit to the calibration task.



3. **Local quantization error as an imperfect proxy:**
The theoretical analysis is grounded in minimizing local quantization error per block.
Yet Figure 2a–b shows that this metric is not strongly predictive of end‑to‑end performance (e.g., green and blue curves show similar local errors but diverge in downstream accuracy).
The limitations of this proxy are not acknowledged or discussed.


4. **Statements in Numerical Analysis inconsistent with earlier arguments:**
The Numerical Analysis section (lines 207–215) states that full Hadamard performs slightly better than block Hadamard for small block sizes.
However, earlier sections emphasize the benefits of blockwise transformations, and Figure 2 seems to show block Hadamard outperforming full Hadamard consistently.
The discrepancy is not explained, and the dependence on block size remains unclear.



5. **Unclear argumentation:**
Section 3.2 (lines 238–240) claims that Section 3.1 “reveals” the set of admissible transformations to be invertible affine functions.
However, Section 3.1 seems instead to motivate the scope of Theorem 3.3 and not define the set of admissible transformations. This connection should be clarified.

I am willing to revise my evaluation whenever the authors address my concerns, especially 1. and 2.

## Minor Issues

* The **Background and Notation** section could be shortened or removed, as much of the content is standard.

* Slight inconsistency: $x$ is described as both a vector and a tensor across Sections 2 and 2.1.

* Equation 4 derives results assuming sub‑Gaussian distributions, yet empirical activation distributions in LLMs are often super‑Gaussian with heavy outliers (as shown in Figure 1). The applicability and usefulness of this derivation is not discussed.

* Section 3.2 (lines 238–240) claims that Section 3.1 “reveals” the set of admissible transformations to be invertible affine functions.
However, Section 3.1 seems instead to motivate the scope of Theorem 3.3. This connection should be clarified.

* In Figure 2, the x‑axis uses different scales for block size, complicating comparisons. The last panel (block index for a fixed block size) is also not clearly distinguished.

* In Figure 2, it is unclear if Hadamard refers to the standars (Sylvester) Hadamard or a randomized Hadamard.

* Figure 4 uses colors inconsistently—there is no clear visual distinction between transformed vs. transformed+quantized activations.

* The QR parametrization resembles that in DartQuant [1], which is not cited.

 ### References
\[1\]: Shao, Yuantian, et al. "DartQuant: Efficient Rotational Distribution Calibration for LLM Quantization.”

---

> ### Author Rebuttal · Authors · 2026-03-31
>
> We thank the reviewer for the constructive and insightful feedback. We are encouraged that the reviewer appreciated the effectiveness of LATMiX and the experimental evaluations. Missing references (DartQuant), relevant discussions, and clarifications will be added to the next version of the paper. Additional results can be found in this anonymous [[LINK]](https://anonymous.4open.science/r/latmix-rebuttal-1).
>
> - **Computational cost & transformations:** LATMiX uses two affine transformations ($T_1$, $T_2$) that can be folded into adjacent linear layers, and $T_3$, a **Hadamard** matrix, which indeed admits an efficient implementation with optimized kernels. To clarify, $T_3$ is **not** a general dense matrix; we use exactly the same transformation for $T_3$ as in previous studies (e.g., MR-GPTQ, SpinQuant). To assess the practical cost, we benchmarked LATMiX on an Nvidia RTX 6000 PRO using optimized kernels from MR-GPTQ and vLLM (Kwon et al., 2023). We observe comparable throughput across all quantized methods, indicating negligible inference cost at worst in practice.
>   | Batch Size | BF16 | RTN | MR-GPTQ | LATMiX-no-bias | LATMiX |
>   |---|---:|---:|---:|---:|---:|
>   | 512 | 3341 | 5797 | 5677 | 5860 | 5533 |
>   | 1024 | 3730 | 9699 | 9658 | 9866 | 9699 |
>   | 2048 | 3595 | 9858 | 9862 | 9841 | 9869 |
>   | 4096 | 3524 | 9866 | 9894 | 9868 | 9950 |
> - **Computational equivalence:** Thank you for this insightful comment. We first note that LATMiX’s zero-shot accuracy clearly improves over existing baselines and, in some cases, remains close to FP model performance. This suggests that harmful overfitting does not occur in practice, since calibration uses only the WikiText2 training set. Furthermore, in response to this concern, we added the following additional experiments showing that LATMiX is robust to the specific calibration set and its size:
>  1. BF16 performance with the learned transformations under MXFP4 — please see our response to Reviewer bwAi. This result indicates that the transformations preserve the network behavior well.
>   2. Zero-shot accuracy when varying the number of calibration samples selected randomly from the WikiText2 training set (Results on WikiText2 testset are in Table 1 at the anonymous link):
>      | Set size | Llama-8B | Qwen-8B |
>      |---|---:|---:|
>      | 1 | 69.11 | 66.33 |
>      | 64 | 70.32 | 68.16 |
>      | 128 | 70.31 | 68.12 |
>      | 256 | 70.51 | 68.46 |
>      | 512 | 71.99 | 69.68 |
>   3. Mean accuracy and std of LATMiX on zero-shot benchmarks over $5$ random subsets of the training data (per-task results are in Table 6 at the anonymous link):
>      | Metric | Llama-8B | Qwen-8B |
>      |---|---:|---:|
>      | Avg. Acc. | 70.25 ± 0.24 | 69.68 ± 0.12 |
>      | Avg. Rec. | 96.01 ± 0.43 | 97.77 ± 0.14 |
> - **Local error metric:** We agree that local quantization error is an imperfect proxy for downstream performance. We use it primarily as a theoretical tool, since it enables a principled analysis of MX quantization error and motivates the design of LATMiX (which, by the way, shows the best performance both according to the local loss and on downstream tasks); similar local metrics are also common in many prior quantization works (e.g., AdaRound, QDrop) due to their tractability and layer-wise insight. That said, these metrics have known limitations, as they generally rely on assumptions such as zero expected task-loss gradient, second-order loss approximation, and layer independence. We will revise the paper to discuss these limitations explicitly including relevant related literature.
> - **Single transform ablation:** The table below shows the benefit of each transformation on PPL.  From the results, we can see the benefit of applying all $3$ transformations, as removing any of them degrades the PPL.
>   | Model | All | No $T_3$ | No $T_1$ | No $T_2$ |
>   |---|---:|---:|---:|---:|
>   | Llama-8B | 7.97 | 8.14 | 8.17 | 8.07 |
> | Qwen-1.7B | 15.72 | 15.87 | 17.38 | 18.16 |
>   | Qwen-8B | 9.38 | 9.54 | 9.60 | 9.47 |
>  - **Inconsistency in Hadamard analysis:** We thank the reviewer for catching our writing mistake which will be corrected (Block Hadamard is preferred over full matrices).
> - **Connection between Sec. 3.2 and Sec. 3.1:** Theorem 3.3 justifies the importance of learning the transformations under MX quantization, and the numerical analysis which accompanies it shows the importance of taking the rich class of invertible affine transformations. Both of which affect the design of LATMiX presented in Section 3.2. We will edit the wording to better accommodate the connection between these sections.
> - **Sub‑Gaussian assumptions:**  We now generalized our theorem to the broader class of $\psi_{\alpha}$​ distributions, which includes heavier-tailed regimes with tails of the form $\exp(-c t^{\alpha})$. In particular,  $\alpha=2$ recovers the sub-Gaussian case, while $\alpha=1$ corresponds to sub-exponential distributions. See full proof at the anonymous link.

---

> > ### Author Rebuttal · Reviewer_Tfqu · 2026-04-02
> >
> > I appreciate the extensive results that the authors have included in the ablation studies. My main concerns about runtime computation and calibration overfitting have been addressed.
> >
> > I acknowledge the relevance of the included ablation regarding the effect of excluding each one of the transforms. However my question were referring specifically to
> >    * Why is $T_3^{-1}$ not fused into the weights of the linear layer according to Figure 4? How is the (inverse) floating point Hadamard matmul applied to the quantized activations and weights (Figure 4c on the right)? Is this an error in the image or am I missing something here? Overall I encourage the authors to better clarify the definition of $T_3$ in the main text.
> >    * How much does the model performance improve when comparing non orthogonal $T_1$ and $T_2$ and orthogonal $T_1$ and $T_2$? In the latter case, the computational invariance constraint is satisfied, while in the former (as in the proposed method) it is not. A more targeted ablation would clarify the specific advantages measured by violating the computational invariance constraint.
> >
> > Since my concerns have been partially addressed, I will increase my score accordingly

---

> > > ### Author Response · Authors · 2026-04-03
> > >
> > > Thank you for the engagement and for increasing the score. We are happy that we addressed the main concerns of the reviewer. The reviewer’s feedback is highly valuable in shaping and making our paper better. Regarding the follow-up questions:
> > >
> > > **1. $T_3^{-1}$ folding** - Thank you for spotting an ambiguity in Figure 4. We agree that it is misleading. $T_3^{-1}$ is fused into the weights of the following linear layer, $W_{down}$ in the figure. So it is an offline transformation. $T_3$ is an online transformation which is not fused to the weights. Namely, it cannot be absorbed into a network’s linear layer. We followed the exact setup of SpinQuant and MR-GPTQ for $T_3$ and its inverse. We will add this clarification to the main text of the paper to prevent any confusion regarding $T_3$.
> > >
> > > **2. Orthogonal $T_1$ and $T_2$** - Thank you for further clarifying your point regarding using orthogonal $T_1$ and $T_2$. First we kindly would like to point the reviewer to our response to Reviewer bwAi (please see our  “Effect of transforms without quantization” comment) which clearly shows that although the computational invariance constraint is not strictly satisfied, the network behaviour remains the same. We witness similar test results before and after integrating the learned affine transformations (under the quantized model) in the original FP16 model.
> > >
> > > Regarding orthogonal $T_1$ and $T_2$. Note that we initialize the training using orthogonal matrices, so if it wasn’t useful to transition to a better location, in terms of the distillation loss, then both transformations would have stayed orthogonal, which is not the case according to Figure 3 in the paper. Furthermore, in the paper, it can be seen that LATMiX is consistently preferred over using orthogonal matrices (ablations in Table 2) and rotation matrices (comparison to SpinQuant in Tables 1 and 3) in almost all cases, both in zero-shot accuracy and test perplexity.
> > >
> > > In addition, in response to reviewer bwAi we added results for orthogonal transformations with bias (please see our “Rotation with bias” comment), showing favourable test perplexity for general invertible matrices. We now add here a similar comparison (LATMiX vs orthogonal $T_1$ and $T_2$ with bias) on zero-shot benchmarks, arriving at the same conclusion.
> > >
> > >  | Method | Llama-1B | Llama-8B | Qwen-1.7B| Qwen-4B
> > >  |---|---:|---:|---:|---:|
> > >  | Orthogonal | 52.05 | 66.75 | 51.91 | 58.14 |
> > >  | Orthogonal with bias | 52.92 | 67.11 | 52.73 | 58.96 |
> > >  | LATMiX | **54.18** | **68.46** | **56.91** | **63.74** |
> > >
> > > **Overall**, we hope that we addressed the reviewer’s concerns appropriately. In case we misunderstood something or any additional clarifications are needed, we will be more than happy to address them

---

### Official Review · Reviewer_Mb4i · 2026-03-13

**Soundness:** 2
**Presentation:** 3
**Significance:** 2
**Originality:** 2
**Overall Recommendation:** 4
**Confidence:** 4

**Summary:**

This paper studies post-training quantization for LLMs under microscaling formats. The authors argue that prior rotation- or Hadamard-based transformations are suboptimal for MX quantization, and propose LATMIX, which learns general invertible affine transformations parameterized via LU or QR decompositions. These transformations are optimized with a KL-divergence distillation objective to redistribute activation outliers more evenly. The paper further claims that the learned transformations can be folded into existing linear layers, leading to no additional inference overhead.

**Compliance With Llm Reviewing Policy:**

Affirmed.

**Final Justification:**

Most of my experimental concerns have been addressed. My remaining concern is about the novelty of the method, as affine transformations have already been used in previous works.

**Key Questions For Authors:**

1. Limited novelty: The use of learnable affine transformations is not entirely new, as related ideas have already appeared in methods such as AffineQuant and FlatQuant.
2. Limited evidence of scalability: A central motivation of LLM quantization is to reduce the memory and inference cost of large models. However, the largest models evaluated here are Llama 3.1 8B and Qwen3 8B. This leaves the practical relevance to 70B-scale or larger models unclear. Since LATMIX requires 1000 to 2500 steps of gradient-based optimization, its calibration-time memory and compute overhead may become substantial at larger scales, but the paper does not provide evidence on this point.
3. Insufficient support for the zero-overhead claim: Appendix C argues that folding the learned transformations into QKV and MLP projection layers leads to zero inference cost. However, in practice, modern LLM inference depends heavily on highly optimized fused kernels and memory layouts. It is not obvious that incorporating general non-orthogonal affine transformations and bias terms can preserve the same end-to-end efficiency. Without actual latency or throughput measurements, the zero-cost claim is not fully substantiated.
4. Modest empirical gains relative to method complexity: The improvements reported in Tables 1 and 3 are generally moderate. For example, on WikiText2, Llama3.2-3B improves from 9.03 under MR-GPTQ to 8.70 under LATMIX. While this is a meaningful gain, it is less clear whether it justifies the added calibration complexity, especially in a PTQ setting where simplicity and efficiency are often major considerations.

**Questions**

1. Please provide end-to-end latency and throughput measurements on real hardware using a standard inference stack to support the claim of zero additional inference overhead.
2. Please provide evidence on scalability, including evaluation on a 70B-scale model if available, as well as the peak calibration memory required by LATMIX.

**Limitations:**

yes

**Strengths And Weaknesses:**

1. Solid theoretical motivation: The paper presents a theoretical upper bound for MX quantization error in Theorem 3.3, showing that the error depends not only on the activation distribution but also on the MX block structure. This provides a reasonable theoretical basis for considering more general non-diagonal affine transformations.
2. Careful engineering design: The parameterizations based on LU and QR decompositions, together with the volume-preserving regularization used to maintain invertibility, are technically well designed and implemented.

---

> ### Author Rebuttal · Authors · 2026-03-31
>
> We thank the reviewer for the constructive and insightful feedback. We are encouraged that the reviewer found interest in our theoretical basis and appreciated LATMiX design choices. Please see our responses to the main claims below. Relevant discussions and clarifications will be added to the next version of the paper.
> - **Novelty:** Thank you for this comment. As we see it LATMiX presents several novelties and valuable contributions to this field: First, to the best of our knowledge, no paper presented a theoretical justification for learning the transformations for quantization in general (applying Theorem 3.3 with $B=d$), let alone under MX quantization. Second, while indeed AffineQuant and FlatQuant proposed using affine transformations, the set of admissible functions proposed in these studies is highly restricted. Our parameterizations are novel under the paper context and can span the full class of affine transformations (in the case of QR). Furthermore, other matrix decompositions can be easily plugged in within our framework, opening the door for future research on this topic. Third, we put forward a novel objective, a distillation loss coupled with a volume-preserving regularization, and propose using standard optimization techniques to learn the model parameters. We further are the first to show empirically, and here in our comments to Reviewer bwAi, that the requirement of strict functional equivalence is not mandatory to obtain high performance.
> - **Scalability:** Thank you for the suggestion. We addressed this point and presented results on larger models - Qwen3-14B and Qwen3-32B in a response to reviewer bwAi. The results for these models show that LATMiX outperforms all baselines, reaching near FP performance in Qwen3-32B. This supports the potential of LATMiX to scale for larger models while maintaining the increase in performance.
> - **Calibration-time memory and compute overhead:** Following the reviewer's suggestion, we measured LATMiX and SpinQuant optimization time. Both methods are comparable, taking approximately two hours to train Qwen3-14B on 4 A100 GPUs. Hence, LATMiX is fully comparable with popular approaches that perform calibration. Moreover, we would like to emphasise that this is a single-time training on WikiText data which is then used for **all** datasets. Regarding the peak memory usage, during calibration LATMiX is largely determined by the choice of batch size and sequence length. In our experiments, we target the largest batch size that fits within the available GPU memory so the training will be faster, as detailed in Section D.1 in the paper. We used ~80GB GPUs in our experiments, but smaller ones can be used with smaller batch sizes or sequence lengths.
> - **Insufficient support for the zero-overhead claim:**  We thank the reviewer for this insightful comment. We agree that, in modern LLM inference, end-to-end efficiency depends heavily on optimized fused kernels and memory layouts, and that theoretical foldability does not automatically guarantee zero practical overhead. To address this concern, we conducted end-to-end throughput measurements in a realistic deployment setting; the corresponding results and detailed discussion are provided in our response to Review Tfqu. The experiment shows that LATMiX’s folded transformations bear negligible overhead in practice at the worst case with or without the biases. In the revised version, we will replace the claim with “negligible overhead” and include the hardware benchmarking results.
> - **Empirical gains relative to method complexity:** In our view, LATMiX is relatively simple as it does not require specialized optimization on manifolds, similarly to SpinQuant and follow-ups, nor does it require special kernels such as FlatQuant. In LATMiX, free-form parameters are defined which parametrize common matrix decompositions, and they are learned using standard DL losses and optimization procedures. So, in terms of method complexity, we believe that LATMiX is favourable compared to other learnable approaches. Empirically, on zero-shot tasks, LATMiX outperforms all baseline methods in 15/16 comparisons with an average improvement in recovery of 2% for MXFP4 quantization, compared to the **best** methods in each comparison. Regarding the perplexity, indeed on some models the improvements are marginal, but on others the gap is meaningful (most prominently, Qwen3-1.7B: 16.88 vs 19.21). Furthermore, on Qwen models LATMiX almost reaches FP16 performance and even improves it. Lastly, we comment that the perplexity of LATMiX can further be improved by using a standard cross-entropy loss as seen in Table 5 in Appendix E.3. However, as we put more emphasis on zero-shot tasks and to maintain a consistent approach, for this task we used the propped distillation loss instead.

---

> > ### Author Rebuttal · Reviewer_Mb4i · 2026-04-04
> >
> > Thanks for your rebuttal.
> >
> > Most of my concerns have been addressed, and I will raise my score. My remaining concern is about the novelty of the work.

---

> > > ### Author Response · Authors · 2026-04-04
> > >
> > > We thank the reviewer for the feedback, which helped improve our paper, and for carefully reviewing our rebuttal. We greatly appreciate the recognition that most of the reviewer’s concerns have been addressed, as well as the positive score.

---

### Decision · Program_Chairs · 2026-04-30

**Decision:**

Accept (regular)

**Comment:**

Initially, reviewers raised some concerns towards the paper and the rebuttal partially resolved them in the first round. After the second round of discussion, I think the authors managed to convince the reviewers. But it is strongly recommended that the authors include the supplementary experiments/figures into the paper in the next revision.